# Pan-serotype dengue virus inhibitor JNJ-A07 targets NS4A-2K-NS4B interaction with NS2B/NS3 and blocks replication organelle formation

Dominik Kiemel[1], Ann-Sophie Helene Kroell [1], Solène Denolly [1], Uta Haselmann[1], Jean-François Bonfanti[2,3], Jose Ignacio Andres[4], Brahma Ghosh [5], Peggy Geluykens[6], Suzanne J. F. Kaptein [7], Lucas Wilken [8], Pietro Scaturro [8], Johan Neyts [7], Marnix Van Loock [9], Olivia Goethals[9] & Ralf Bartenschlager [1,10] ✉

Dengue fever represents a significant medical and socio-economic burden in (sub)tropical regions, yet antivirals for treatment or prophylaxis are lacking. JNJ-A07 was described as highly active against the different genotypes within each serotype of the disease-causing dengue virus (DENV). Based on clustering of resistance mutations it has been assumed to target DENV non-structural protein 4B (NS4B). Using a photoaffinity labeling compound with high structural similarity to JNJ-A07, here we demonstrate binding to NS4B and its precursor NS4A-2K-NS4B. Consistently, we report recruitment of the compound to intracellular sites enriched for these proteins. We further specify the mechanism-of-action of JNJ-A07, which has virtually no effect on viral polyprotein cleavage, but targets the interaction between the NS2B/NS3 protease/helicase complex and the NS4A-2K-NS4B cleavage intermediate. This interaction is functionally linked to de novo formation of vesicle packets (VPs), the sites of DENV RNA replication. JNJ-A07 blocks VPs biogenesis with little effect on established ones. A similar mechanism-of-action was found for another NS4B inhibitor, NITD-688. In summary, we unravel the antiviral mechanism of these NS4B-targeting molecules and show how DENV employs a short-lived cleavage intermediate to carry out an early step of the viral life cycle.

Dengue virus (DENV) is a highly prevalent pathogen of the *Flaviviridae* family to which yellow fever virus (YFV) and Zika virus (ZIKV) belong. These viruses are transmitted to humans through the bite of an infected female mosquito of the genus *Aedes*[1]. The current widespread distribution of DENV in tropical and subtropical latitudes has been accelerated by man-made influences, including spread of the vectors through commodity trade and climate change[2]. Although the majority of the ~100 million annual symptomatic infections cause mild to moderate

dengue fever, the burden of disease is enormous[3]. DENV infections can also have life-threatening outcome due to dengue hemorrhagic fever and dengue shock syndrome[4]. Four serotypes of DENV are known, all of which are pathogenic, with serial heterotypic infections tending to have a more severe course than primary infection, which has been ascribed to antibody-dependent enhancement of infection[5–8].

The genome of DENV is an ~11,000 nucleotides long single strand RNA of positive polarity[9]. It possesses one long open reading frame

flanked on both sides by non-translated regions (NTRs) and encodes a polyprotein, which is co- and post-translationally cleaved into 10 different proteins. Their order in the polyprotein is (from N- to C-terminus): capsid (C), precursor membrane (prM), envelope (E), non-structural protein 1 (NS1), NS2A, NS2B, NS3, NS4A, NS4B and NS5[10]. Polyprotein cleavage follows a preferential order and is mediated by the NS2B/NS3 protease and additional host cell proteases[11–13]. While C, prM and E are major components of the DENV virion, all non-structural proteins are required for viral RNA replication. Amongst these, NS2B is a transmembrane protein that tethers NS3 to ER membranes and activates the serine-type protease residing in the N-terminal NS3 domain while the C-terminal domain is a helicase[14]. NS4A and NS4B are highly hydrophobic transmembrane proteins that can remodel intracellular membranes[10]. NS5 is the RNA-dependent RNA polymerase and is also required for capping and methylation of the viral RNA genome[15–17]. In addition to mature products, cleavage intermediates are detectable in infected cells, most notably an NS4A-2K-NS4B precursor with 2K being a small trans-membrane fragment serving as signal sequence for NS4B[13,18–20]. While processing between NS4A and 2K is mediated by the viral NS2B/NS3 protease, cleavage at the 2K-NS4B site is mediated by the host cell signalase, but requires prior cleavage at the NS4A-2K site[21–23].

NS4B is a multi-pass transmembrane protein that lacks enzymatic activity and is post-translationally N-glycosylated at positions N58 and N62[24]. NS4B can form homodimers, for which the region from residue 125 to the C-terminus is crucial, but also interacts with multiple host cell proteins[25]. These interactions have been linked to evasion from innate immunity and inhibition of cellular processes such as stress-granule formation[26,27]. NS4B interactions with viral proteins have been reported for NS4A and NS3, the latter enhancing NS3 helicase activity[28–31]. While mature NS4B does not interact with NS1, it does so when being part of the metastable NS4A-2K-NS4B cleavage intermediate and this interaction is necessary for viral RNA replication[18]. Upon infection of the host cell, the viral RNA is released into the cytoplasm and directly used for synthesis of the polyprotein at the rough ER. There, most likely by a concerted action of DENV non-structural proteins and several host cell factors, the ER is remodeled to form multiple vesicular invaginations, designated vesicle packets (VPs)[32,33]. These vesicles have an average diameter of 90 nm and are connected to the cytoplasm by an ~11 nm wide pore. Presumably within these vesicles the viral RNA genome is amplified via a negative-strand intermediate with progeny RNA genomes being released through the pore into the cytoplasm. There they are used for the synthesis of new polyproteins or packaged into virus particles[32]. Formation of VPs can be induced by the sole expression of the non-structural proteins independent of RNA replication[34]. A second membrane structure found in mammalian, but not in DENV-infected mosquito cells are convoluted membranes, which are accumulations of ER-derived membranes of poorly defined function[35,36].

We recently described the pan-serotype DENV inhibitor JNJ-A07 that was developed by cell-based phenotypic screening and subsequent medicinal chemistry efforts[37]. JNJ-A07 has nano- to picomolar activity and demonstrated a favorable pharmacokinetic and safety profile in rodents. Notably, in mice, therapeutic effects were found in prophylactic settings, but also when treatment initiation was delayed by several days after infection[37]. Mosnodenvir (a compound from the same chemical space as JNJ-A07; formerly known as JNJ-1802; for structure and synthesis see ref. 38), is also highly effective against infection with either DENV-1 or DENV-2 in non-human primates[39]. Adaptive mutations to both compounds obtained via in vitro resistance selection map to NS4B arguing that NS4B is the target. In addition, two other small molecules, NITD-688 and BDAA inhibiting DENV and YFV replication, respectively, select for resistance-associated mutations in NS4B of either virus[40,41]. JNJ-A07 and mosnodenvir prevent the interaction between the NS2B/NS3 complex and NS4B-containing proteins[37,39]. However, the exact antiviral mechanism remained unclear.

In this work, we provide a detailed analysis showing that JNJ-A07 inhibits predominantly the interaction between NS2B/NS3 and the NS4A-2K-NS4B precursor without impacting cleavage kinetics of the latter. This interaction is functionally linked to the de novo formation of DENV VPs, which are essential for the virus as they house the processes of viral RNA replication. Our results represent a compelling explanation for the mechanism-of-action of JNJ-A07 and possibly other NS4B-targeting compounds.

## Results

### PAL derivatives of JNJ-A07 accumulate intracellularly at sites enriched for DENV NS3 and NS4B

When investigating the mechanism-of-action of replication-blocking DENV compounds such as JNJ-A07, the fundamental limitation in replication-based assays is that all stages of the viral replication cycle, including protein synthesis, polyprotein processing, VPs formation and genome amplification are inseparably linked, making it difficult to identify the precise step affected by the compound. For this reason, we used expression-based systems either by transfection of plasmids encoding individual viral proteins or a recently reported approach designated pIRO-D system, an acronym derived from *plasmid-induced replication organelles of DENV*[34]. In this system, cells stably expressing the RNA polymerase of bacteriophage T7 are transfected with a plasmid encoding the non-structural proteins of the DENV-2 polyprotein (strain Thailand/16681) under control of the T7 promoter. The coding region is 5' flanked by fragments of the DENV-2 capsid sequence and 3' by the 3' NTR, with the viral proteins inducing the formation of bona fide VPs in transfected cells (Supplementary Fig. 1a).

To characterize compound association to DENV NS4B, we conducted imaging-based studies determining colocalization of the compound with viral NS4B and, as reference, NS3. We applied a photoaffinity labeling (PAL) approach using Compound A and Compound B having high structural similarity to JNJ-A07 and being enantiomers of each other (Supplementary Fig. 1b; for synthesis see Supplementary Methods and Supplementary Fig. 2). Compound A is the (R)-enantiomer and about 60-times more antivirally active in DENV-infected cells than the (S)-enantiomer Compound B. These PAL compounds contained a diazirine group for covalent cross-linking to target structures in close proximity by using UV irradiation (365 nm), and a terminal alkyne that allows conjugation with an adapter molecule in a click-chemistry reaction[42]. Huh7/Lunet-T7 cells, having high transfection efficiency and being suitable for imaging analyses[34] were transfected with the plasmid encoding the wildtype (WT) NS1-5 polyprotein (pIRO-D) and treated with PAL compounds or vehicle control. After 4 h, cells were UV irradiated and fixed after an additional 14 h (Fig. 1a). Bound compound was visualized after click-reaction with Picolyl-azide-Cy5.5 (Jena Bioscience) and NS3 and NS4B were detected by immunofluorescence (IF). Both viral proteins showed a high degree of colocalization with labeled Compound A in large puncta possibly corresponding to convoluted membranes and/or VPs, whereas vehicle-treated control cells and mock-transfected cells showed only diffuse background pattern (Fig. 1b). Pearson correlation coefficients between NS4B and the labeled PAL compounds revealed the less active Compound B having significantly lower colocalization than the more active Compound A, but for both labeled compounds, colocalization was well above background (Fig. 1c). To corroborate specificity, we conducted the same assay but using the ZIKV NS1-5 polyprotein (pIRO-Z). Consistent with marginal antiviral activity of JNJ-A07 against ZIKV[37], we detected low-level, albeit significant colocalization of the labeled compound with ZIKV NS4B (Fig. 1d, e). These results argue for specific accumulation of the compound at DENV-induced intracellular structures enriched for NS4B and NS3.

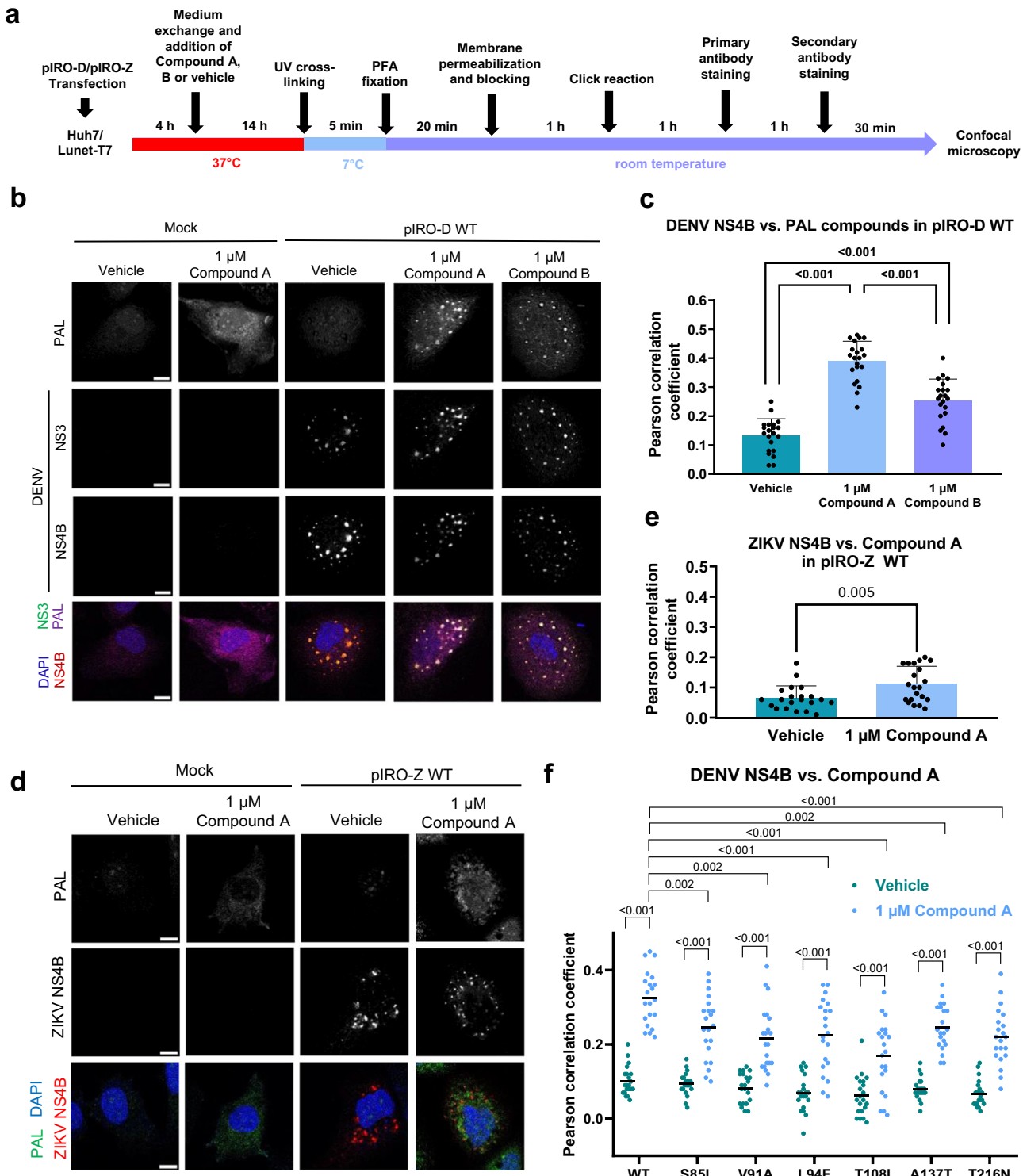

We have earlier reported several NS4B mutations conferring various degrees of resistance against JNJ-A07 and having different effects on viral replication fitness (summarized in Supplementary Fig. 3a)[37]. To correlate resistance with compound binding, we first checked to what extent these JNJ-A07 resistance mutations conferred resistance to Compound A. Although there were some differences, overall the distinction of strong (L94F, V91A, T216N) and weak (S85L, T108I and A137T) resistance mutations was comparable for both compounds (Supplementary Fig. 3a). Next, we analyzed these NS4B mutants using the colocalization assay described above. As for WT, all mutant NS4Bs displayed a dot-like pattern colocalizing with labeled Compound A

(Supplementary Fig. 3b). The lower NS4B signal observed for mutants A137T and T216N relative to WT was probably due to reduced primary antibody binding as the mutations reside in its epitope. A consistently high degree of colocalization between NS3 and the various NS4Bs (Supplementary Fig. 4a) and between labeled Compound A and each of the two viral proteins was found (Fig. 1f and Supplementary Fig. 4b, respectively). Notably, the lower colocalization between Compound A and the two viral proteins, relative to WT, did not correspond to the respective level of Compound A resistance. Thus, while reduced compound binding might contribute to NS4B inhibitor resistance, further so far unknown mechanisms seem to add to resistance.

**Fig. 1 | Compound A specifically localizes to intracellular sites enriched for DENV NS4B and NS3. a** Schematic representation of the workflow to perform click-labeling and immunofluorescence staining on fixed cells that were treated with one of the PAL enantiomers or DMSO control (vehicle). Note that a click reaction with Picolyl-azide-Cy5.5 was performed regardless of treatment condition. **b** Example confocal microscopic images (60x). Cells were transfected with the pIRO-D WT construct encoding the NS1-5 polyprotein or mock transfected and treated with either 1 μM Compound A, 1 μM Compound B or vehicle. Shown are the PAL fluorescence signal, the IF signals of NS3 and NS4B and a merge that includes nuclear DNA stain with DAPI. Scale bar = 10 μm. **c** For each sample transfected with pIRO-D WT, 21 cell profiles (7 from each of the 3 independent experiments) were examined for Pearson correlation coefficients between the PAL and the NS4B signal. Shown is the mean with standard deviation (SD). A one-way ANOVA followed by Tukey's multiple comparisons test was used to determine statistical parameters. **d** Specificity of Compound A colocalization with NS4B was determined using cells transfected with a ZIKV polyprotein encoding construct (pIRO-Z). Example images obtained as described for **b** are shown. Scalebar = 10 μm. **e** Quantification of PAL and ZIKV NS4B signal overlap as described for **c**. Per sample, 21 cell profiles (7 from each of the 3 independent experiments) were analyzed and plotted with mean and SD. The indicated *p*-value resulted from an unpaired two-tailed *t*-test. **f** NS4B mutations conferring various degrees of resistance against JNJ-A07 were examined for their impact on colocalization between NS4B and PAL signals. The corresponding IF panel is shown in Supplementary Fig. 3b. Pearson correlation coefficients of a total of 21 cell profiles are plotted (7 each from three independent experiments). Statistical significance was calculated using a two-way ANOVA test and Šidák's multiple comparisons. ns = non-significant.

## Compound A binds to mature NS4B and the NS4A-2K-NS4B cleavage intermediate

The high spatial proximity of JNJ-A07 with DENV NS4B argued for compound binding to this viral target. To corroborate this assumption and to find out whether the compound binds to mature NS4B and/or the NS4A-2K-NS4B precursor, we established a method that overcomes an observed aggregation of NS4B and its precursor in cell lysates upon addition of the click chemistry reagents. This method utilized the trans-cleavage system in which Huh7/Lunet cells stably expressing T7 RNA polymerase and NS2B-NS3 were transfected with constructs encoding NS4A-2K-NS4B-HA, or 2K-NS4B-HA (2K is necessary for proper membrane insertion of NS4B). Target proteins were immobilized on HA-agarose beads prior to the click reaction with Picolyl-azide-Cy5.5 (Fig. 2a). Although some aggregation could still be observed in these samples, strong Cy5.5 labeling was detected on both NS4B and NS4B-2K-NS4B (Fig. 2b [Input]; Fig. 2c lanes 4&6 [IP]), whereas those treated with DMSO showed only moderate, nonspecific labeling (Fig. 2c lanes 3&5). No labeling was found with HA-tagged Calnexin, an integral ER-resident protein known to colocalize with DENV NS4B[43] (Fig. 2c lanes 7&8). For all analyzed NS4B species, quantified binding efficiencies were significantly increased over Calnexin-HA, indicative of specific compound binding (Fig. 2d). Whether Compound A has a higher affinity to mature NS4B or NS4A-2K-NS4B is unclear, because these comparisons did not reach statistical significance.

## No detectable effect of JNJ-A07 on the cleavage kinetics of NS4A-2K-NS4B in the polyprotein context

In our previous work, we reported that JNJ-A07 might alter the cleavage kinetics of the NS4A-2K-NS4B precursor as deduced from a dose-dependent decrease in the 2K-NS4B cleavage intermediate[37]. This observation was made in the trans-cleavage system (Fig. 2a). To determine cleavage kinetics in a polyprotein context, we utilized Huh7/Lunet-T7 cells transfected with a pIRO-D construct encoding the NS1-5 polyprotein with a replication-competent HA-tagged NS4B[21] (NS4B-HA*) (Fig. 3a) and conducted pulse-chase experiments in presence or absence of JNJ-A07. Proteins were radio-labeled with [$^{35}$S] cysteine and methionine for 20 min, followed by incubation in non-radioactive medium for different time spans. Fully cleaved NS4B-HA* and corresponding cleavage intermediates were enriched by pulldown and proteins were analyzed by SDS-PAGE and autoradiography. For both JNJ-A07- and vehicle-treated control cells we observed a rapid and complete conversion of the NS4A-2K-NS4B precursor to the 2K-NS4B intermediate that was converted into mature NS4B (Fig. 3b). Quantification of autoradiographs revealed very similar cleavage kinetics of NS4A-2K-NS4B in JNJ-A07-treated and control cells (Fig. 3c) with NS4A-2K-NS4B precursor half-lives of $13.6 \pm 2.0$ min and $15.4 \pm 2.6$ min, respectively (Fig. 3d). This result suggests that the NS4B inhibitor JNJ-A07 has no significant effect on processing kinetics of NS4B containing polyprotein species.

## Impact of JNJ-A07 on the interaction between NS2B/NS3 and the NS4A-2K-NS4B precursor

Given the various NS4B-containing species (NS4A-2K-NS4B, 2K-NS4B and NS4B), we wondered whether NS2B/NS3 interacts with all these proteins and if so, which of these interactions would be affected by JNJ-A07. To this end, we applied the trans-cleavage system using both Huh7 cells (employed in previous studies of the interaction between NS4B-containing species and NS2B/NS3[21,37]) and its subclone Huh7/Lunet (for which the pIRO system has been established and which are particularly suitable for imaging[34]), each expressing the T7 RNA polymerase and NS2B/NS3. Cells were transfected with constructs encoding NS4A-2K-NS4B-HA or 2K-NS4B-HA (Fig. 4a); analogous constructs with the Q134A mutation in NS4B, abolishing NS2B/NS3 interaction with NS4B[21], served as control. Samples were processed as summarized in Fig. 4a. Although expression levels were higher in Huh7/Lunet-T7 cells, in both cell lines the expected triad of NS4B, 2K-NS4B, and NS4A-2K-NS4B was detected in the input, the latter being most abundant (Fig. 4b lanes 2&9, c). Co-precipitation of NS3 was drastically reduced upon treatment with 35 nM JNJ-A07 (Fig. 4b lanes 3&10, c), which is ~45-fold above the EC$_{50}$ determined in virus-based replication assay, or with the Q134A non-binder mutant (Fig. 4b lanes 4&11, c). In cells expressing 2K-NS4B, mature NS4B was detected in the input along with a ladder of high molecular weight bands that may represent oligomers, while uncleaved 2K-NS4B was virtually absent (Fig. 4b lanes 5&12, c). Importantly, no co-precipitation of NS3 with 2K-NS4B or NS4B was detected. These results suggest that JNJ-A07 blocks the interaction of NS2B/NS3 with the NS4A-2K-NS4B precursor and not, or to a much lesser extent, with mature NS4B.

To corroborate this conclusion, we used two additional approaches in which the NS4A-2K-NS4B precursor is not cleaved, while the interaction with the NS2B/NS3 complex can still form (Supplementary Fig. 5a). First, we used an NS4A-2K-NS4B mutant in which the protease cleavage site (CS) at the NS4A-2K junction was blocked by an isoleucine substitution at the P1' position (mutant CS P1'Ile). Second, we co-expressed WT NS4A-2K-NS4B along with a protease-inactive NS2B/NS3 mutant (the active site serine residue S135 was replaced by alanine). In both conditions, a stable NS4A-2K-NS4B precursor, but very little or no 2K-NS4B and NS4B were detectable (Fig. 4d, quantified in Supplementary Fig. 5b). Importantly, JNJ-A07 reduced the interaction between NS4A-2K-NS4B and NS2B/NS3 in a dose-dependent manner with EC$_{50}$ values in the single digit nanomolar range (3.3 nM in the case of CS P1'Ile and 4.1 nM for protease-inactive NS2B/NS3 mutant). This value is very close to the EC$_{50}$ concentration of 6.0 nM that we reported earlier when using cells with the active NS2B/NS3 protease complex co-expressed with NS4A-2K-NS4B (Fig. 4e), supporting the conclusion that JNJ-A07 primarily affects NS4A-2K-NS4B precursor interaction with NS2B/NS3.

Next, we determined whether the observed interaction with NS3 is a precursor-intrinsic property or mediated by the NS4A moiety by using constructs encoding N-terminally HA-tagged NS4A (Fig. 5a). In cells expressing HA-NS4A, we did not detect NS2B/NS3 co-precipitation

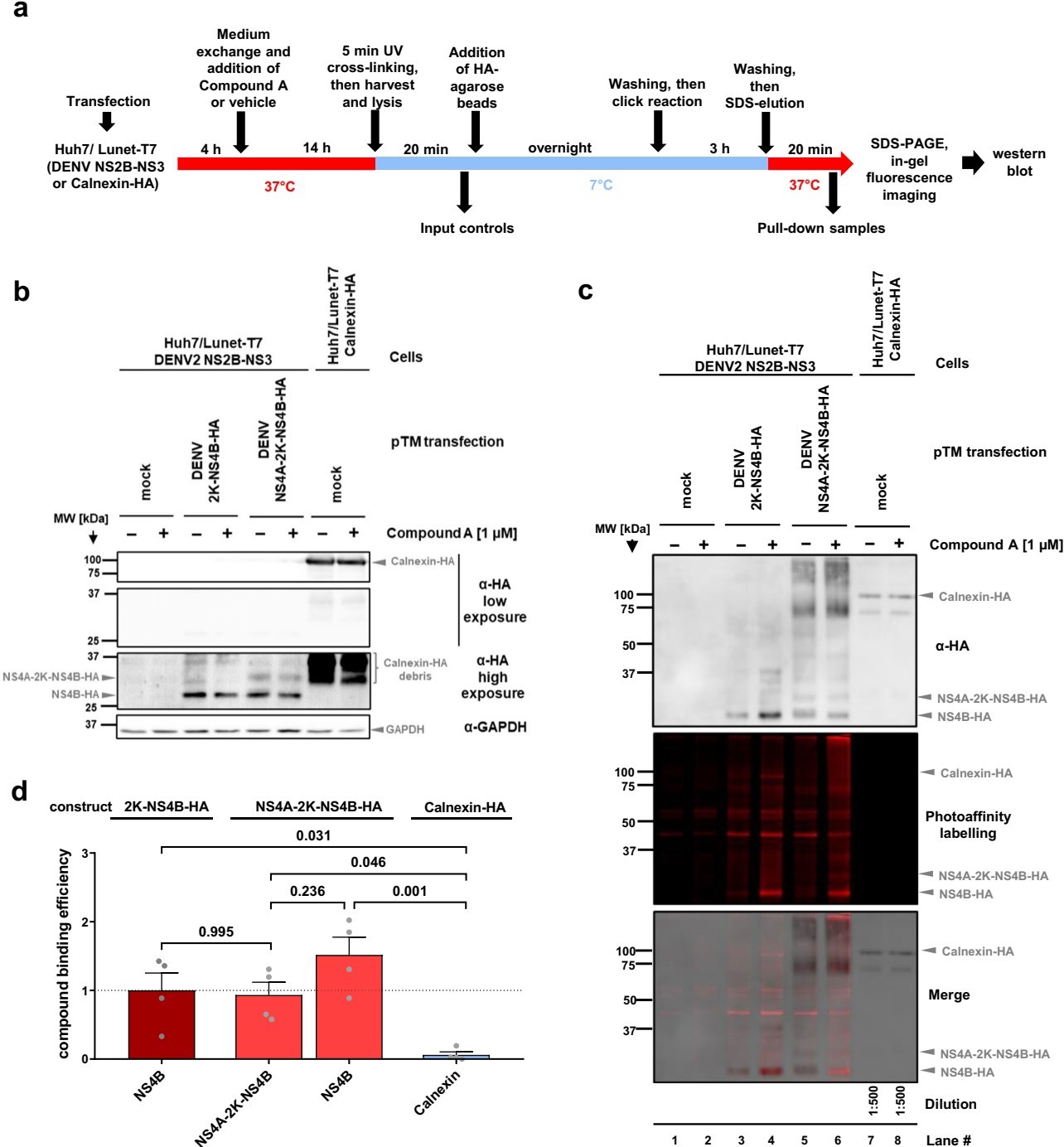

**Fig. 2 | Compound A binds to mature NS4B and the NS4A-2K-NS4B precursor.**
**a** Schematic representation of the workflow. Note that although cells were treated with either Compound A or DMSO control (vehicle), the click reaction was performed on the beads with Picolyl-azide-Cy5.5 azide in both cases. Also note that different subclones of Huh7/Lunet-T7 cells were used: naïve ones in which 2K-NS4B-HA or NS4A-2K-NS4B-HA were transiently expressed and a subclone stably expressing Calnexin-HA. **b** Representative western blot showing the input controls. Due to its stable expression, the signal of Calnexin-HA is stronger than the ones of the transiently expressed viral proteins. Mock samples were derived from Huh7/Lunet-T7 NS2B-NS3 cells treated with transfection reagent only. **c** Representative images to analyze compound binding to individual proteins. Pull-down samples were loaded onto a gel, which was analyzed for Cy5.5 fluorescence signal using a LICOR Odyssey imaging system (middle plane). Thereafter, proteins were analyzed by HA-specific western blot (top panel). An overlay of both images is shown at the

bottom. Mock transfected samples in lanes 1 and 2 reflect the unspecific background in the absence of HA-tagged proteins. For each construct, there is a Compound A-treated sample and a vehicle-treated sample to distinguish between non-specific labeling and specific binding to Compound A. Owing to high expression level, Calnexin-HA samples were diluted 1:500 prior to loading to reach western blot signal intensities in a linear range comparable with the other samples. **d** Quantification of compound binding efficiency; shown is the ratio of the fluorescence signals (reflecting Compound A bound proteins) to the corresponding HA western blot signal (reflecting the target protein). Non-specific compound labelling was subtracted by using the analogous ratio of the individual vehicle control. Bar graph shows individual data points, as well as mean and standard error of the mean (SEM; $n = 4$). Indicated $p$-values were calculated using one-way ANOVA with subsequent Tukey's multiple comparisons test.

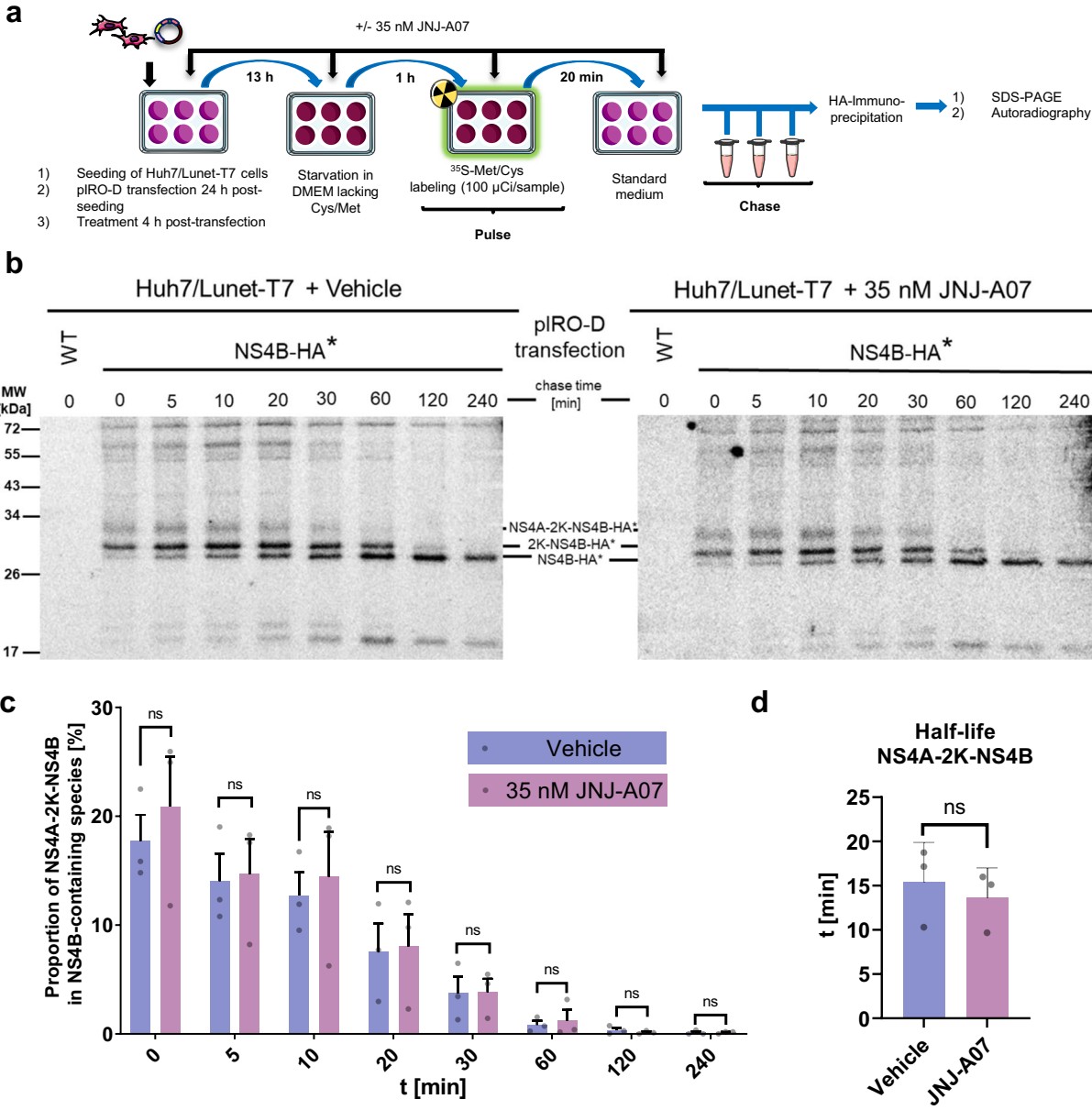

**Fig. 3 | Cleavage kinetics of the NS4A-2K-NS4B precursor in the polyprotein context is not affected by JNJ-A07. a** Experimental workflow. Huh7/Lunet-T7 cells were transfected with a pIRO-D construct corresponding to the NS1-5 polyprotein and containing an HA-affinity tag in the N-terminal region of NS4B (HA*) after the 2 K peptide[21]. After 60 minutes starvation in methionine and cysteine-free medium, cells were incubated in radioactive medium for 20 min (pulse) followed by incubation for various time spans in non-radioactive medium (chase). Proteins were enriched by HA-IP and after SDS-PAGE analyzed by autoradiography. **b** A representative autoradiogram from a total of 3 independent experiments is shown. Numbers above the lanes refer to chase time (minutes). **c** Signal intensities of NS4B-specific bands were quantified and adjusted to cysteine and methionine residues contained therein. The proportion of NS4A-2K-NS4B amongst all NS4B-containing species is plotted with mean and SEM ($n = 3$). No statistically significant differences were found between JNJ-A07 and vehicle control-treated samples for any time point as determined by a Sidak's multiple comparisons test following a two-way ANOVA. **d** "One phase exponential decay"-curves were fitted to the result of each experiment ($n = 3$) to obtain three NS4A-2K-NS4B half-life values for both the JNJ-A07 and vehicle control-treated setup that are plotted here with mean and SEM. The difference was not significant as determined by a paired two-tailed two sample $t$-test.

regardless of whether JNJ-A07 was added or not (Fig. 5b lanes 4&5, c). In contrast, when the N-terminally HA-tagged NS4A-2K-NS4B precursor was co-expressed with NS2B/NS3, the compound-sensitive NS2B/NS3 interaction was reproduced (Fig. 5b lanes 6&7, c). Furthermore, JNJ-A07 did not induce cleavage between 2K and NS4B, because we did not detect NS4A-2K in samples isolated from compound-treated cells, while a consistent accumulation of uncleaved precursor along with reduced amounts of HA-NS4A was detected, which was best visible in the IP samples (Fig. 5d, lanes 3&4).

Results obtained so far were generated with the trans-cleavage system. However, in infected cells the viral protease is part of the polyprotein substrate and therefore, we re-analyzed the impact of JNJ-A07 on precursor–protease interaction in the polyprotein (pIRO-D) system. To this end, we conducted pull-down experiments using lysates of Huh7/Lunet-T7 cells expressing a polyprotein containing a replication-competent HA-tag residing in NS4B and which had been treated for 18 h with JNJ-A07. Consistent with the trans-cleavage system, we found that interactions between NS2B/NS3 and NS4B-

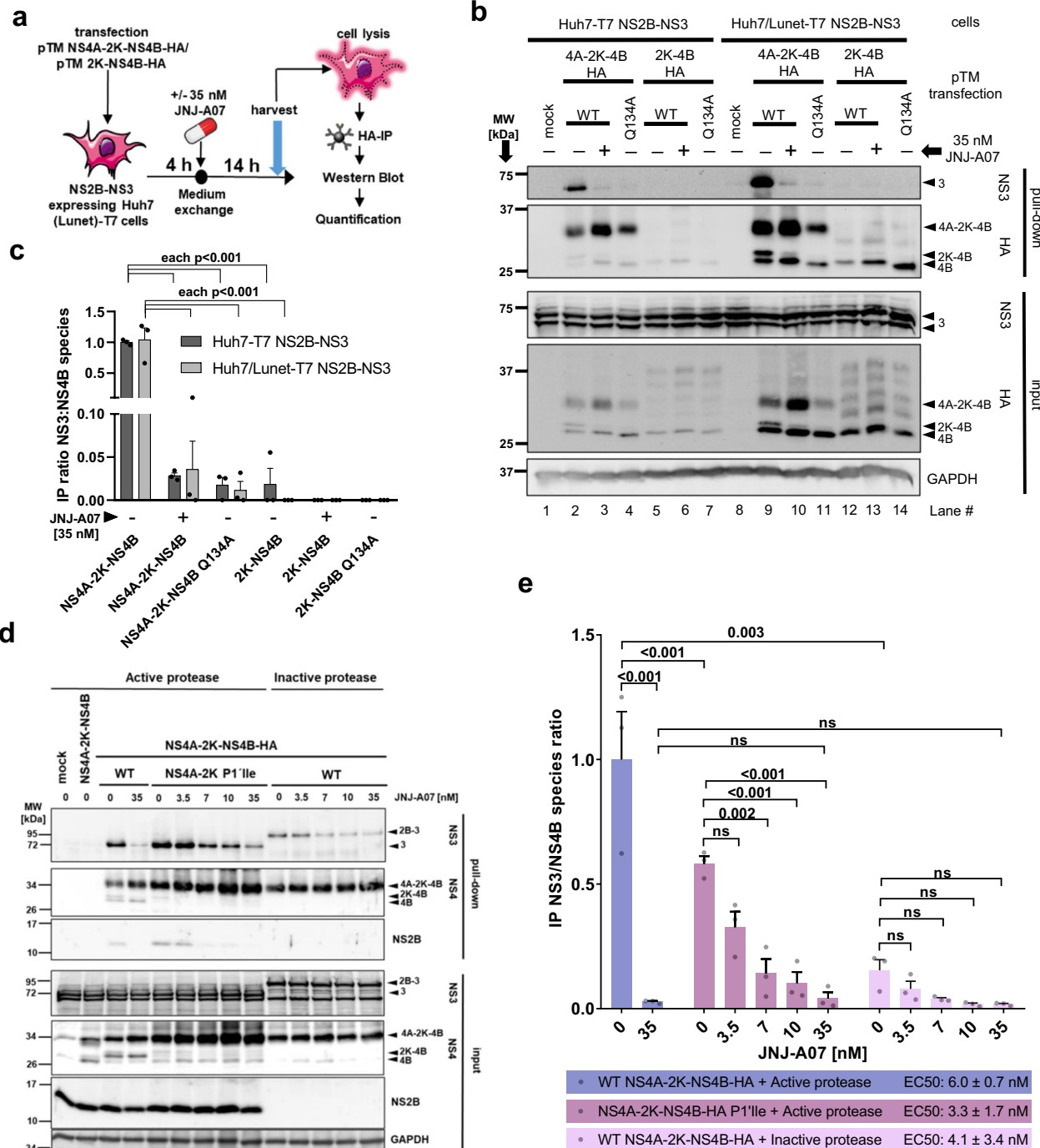

**Fig. 4 | NS2B/NS3 interacts with NS4A-2K-NS4B, but not with mature NS4B, and this interaction is prevented by JNJ-A07 in the trans-cleavage system.**
**a** Experimental approach: Huh7-T7 or Huh7/Lunet-T7 cells, each stably expressing the T7 RNA polymerase and DENV NS2B/NS3 were transfected with T7-based expression plasmids encoding either NS4A-2K-NS4B-HA or 2K-NS4B-HA with NS4B corresponding to the wild-type (WT) or containing the NS3-nonbinder mutation Q134A. After 4 h, cells were treated with JNJ-A07 or vehicle, collected 14 h thereafter and lysates were used for HA-specific pull-down. **b** Side-by-side comparison of NS3 co-precipitation with either NS4A-2K-NS4B-HA or 2K-NS4B-HA. Eluates were analyzed by western blot along with 25% of the input. A representative experiment is shown. Arrowheads indicate named DENV non-structural proteins, whereby the usual "NS" nomenclature has been omitted here for reasons of space. **c** The ratios of NS3 to NS4B species (NS4A-2K-NS4B, 2K-NS4B and NS4B) were determined by

quantifying 3 independent experiments and are plotted with mean and SEM. A two-way ANOVA preceded by Dunnet´s test was conducted to determine given p-values. **d** Interaction between NS2B/NS3 and stabilized NS4A-2K-NS4B precursors. Huh7-T7 cells expressing either the active DENV protease complex or an inactive version thereof (NS3 mutant S135A) were transfected with indicated expression constructs. Shown is a representative western blot result (*n* = 3). **e** IP ratios of HA-precipitated NS4A-2K-NS4B and co-captured NS3 were calculated from 3 independent experiments and normalized to vehicle-treated WT NS4A-2K-NS4B co-expressed with the active protease. Mean and SEM are shown; p-values were calculated by Šidák's multiple comparisons test following one-way ANOVA (ns = non-significant). EC$_{50}$ values given in the bottom lines were calculated by fitting dose-response curves or taken from Kaptein et al., 2021[37] (for "WT NS4A-2K-NS4B-HA + Active protease" shown here as reference).

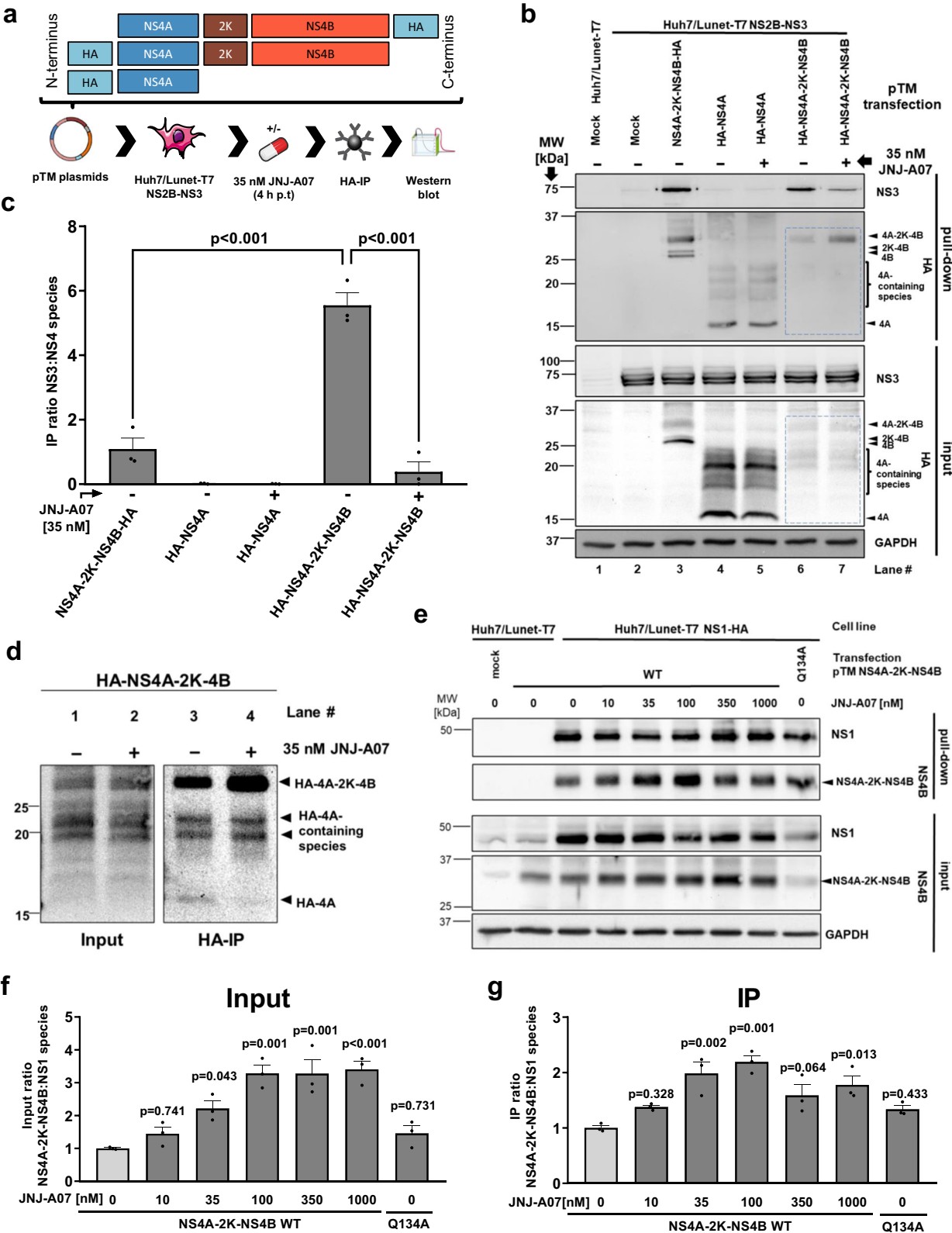

containing species formed, but addition of JNJ-A07 did not result in a loss of NS3 co-precipitation with the precursor (Supplementary Fig. 5c, d). This result suggests that additional protein–protein interactions not occurring in the trans-cleavage system might stabilize and thus, mask the impact of JNJ-A07 on the association of NS2B/NS3 with the NS4A-2K-NS4B precursor.

## JNJ-A07 does not affect the interaction between NS1 and the NS4A-2K-NS4B precursor

We showed recently that the NS4A-2K-NS4B precursor interacts with NS1[18]. Therefore, we wondered whether JNJ-A07 might also block this interaction and transfected the plasmid encoding NS4A-2K-NS4B into Huh7/Lunet-T7 cells stably expressing NS1-HA. This tagged NS1 is fully

**Fig. 5 | JNJ-A07 neither affects the cleavage order of NS4A-2K-NS4B, nor its interaction with NS1. a** Constructs specified on the top were transfected into Huh7/Lunet-T7 cells stably expressing NS2B-NS3. 4 h post-transfection, cells were treated with either 35 nM JNJ-A07 or vehicle. After another 14 h, samples were harvested and used for HA-specific pull-down. Captured proteins were analyzed by western blot. The C-terminally HA-tagged NS4A-2K-NS4B precursor was included as reference. **b** Representative western blot showing HA-pull-down samples and total lysates (25% of input). GAPDH served as loading control for cell lysates. The section highlighted in panel d is indicated with a blue dashed line box. **c** Bar graph showing the ratio of NS3 to NS4 species (mean and SEM) as determined by quantification of independent western blots ($n = 3$). *P*-values were calculated using one-way ANOVA with subsequent Šidák's multiple comparisons test. **d** Sections of the western blots

highlighted in **b** adjusted for brightness and contrast to highlight NS4A species in pTM HA-NS4B-2K-NS4B transfected samples. **e** Huh7/Lunet-T7 cells stably expressing NS1-HA were transfected with constructs encoding NS4A-2K-NS4B. After 4 h, cells were treated with given concentrations of JNJ-A07 or vehicle, collected 14 h thereafter and lysates were used for HA-specific pull-down. Captured protein complexes were analyzed by western blot. A representative example of 3 independent experiments is shown. GAPDH served as loading control for cell lysates (input). **f, g** Ratios of NS1 to NS4A-2K-NS4B in input and pull-down samples were calculated with mean and SEM by analyzing signal intensities in the 3 independent experiments. For statistical analysis, one-way ANOVA plus post-hoc test was performed using Dunnett's correction to account for multiple testing.

functional as it rescues the replication of a DENV NS1 deletion mutant unable to replicate on its own[18]. Lysates of cells cultured in the presence of various concentrations of JNJ-A07 for 14 h were evaluated either directly or used for HA-specific pull-down and proteins were analyzed by quantitative western blot (Fig. 5e). Up to 100 nM JNJ-A07, a dose-dependent increase in the ratio of NS4A-2K-NS4B to NS1 was observed in the cell lysates (Fig. 5f), presumably due to stabilization of the precursor in this system. An analogous increase was quantified in the pull-down samples (Fig. 5g). Importantly, JNJ-A07 did not reduce the NS4A-2K-NS4B interaction with NS1 up to compound concentrations that were around 165-fold higher than that required to prevent precursor interaction with NS2B/NS3 ($EC_{50}$ 6.0 nM). The same result was found when we analyzed precursor interaction with NS1 in the polyprotein (pIRO-D) system (Supplementary Fig. 5c). Taken together, these results show that JNJ-A07 has no impact on the association of NS4A-2K-NS4B with NS1. In addition, the data suggest that this compound does not induce global misfolding of NS4A-2K-NS4B, because interaction with NS1 is retained, arguing for a more selective block of precursor association with NS2B/NS3.

## JNJ-A07 blocks de novo formation of DENV replication organelles

JNJ-A07 blocks DENV RNA replication at a very early stage of the viral replication cycle, i.e. after RNA genome release into the cytoplasm and prior to virion assembly[37]. Moreover, we and others found that NS4B and NS4A play very critical roles in membrane remodeling[10]. Therefore, we assumed that JNJ-A07 might affect the formation of VPs. We employed the pIRO-D system and transfected Huh7/Lunet-T7 cells with constructs encoding for the WT NS1-5 polyprotein or variants thereof containing JNJ-A07 resistance mutations in NS4B and treated the cells with 35 nM JNJ-A07 or DMSO (Fig. 6a). Cells were harvested 18 h after transfection and used to determine transfection efficiency by IF, polyprotein processing by western blot, and VPs formation by transmission electron microscopy (TEM). On average, transfection efficiencies were in the range of 50% for most of the samples, which is sufficiently high for TEM without the need for correlative microscopy approaches (Fig. 6b). Polyprotein processing was neither affected by the NS4B resistance mutations, nor by JNJ-A07 treatment (Fig. 6c). However, the cleavage pattern of NS4B-containing species was clearly different from the one observed in the NS4A-2K-NS4B trans-cleavage system as we almost exclusively detected fully processed NS4B in the case of the NS1-5 polyprotein (compare Fig. 4b with 6c, respectively). This is most likely due to high cleavage efficiency in this system, consistent with our pulse-chase experiments. Regardless of the NS4B mutation and treatment, membrane rearrangements were detected in all TEM samples (Fig. 6d, e). These rearrangements comprise convoluted membranes (CMs) and VPs, the latter consisting of frequently linear arrays of single-membrane vesicles that in TEM appear surrounded by a second ER membrane (Fig. 6d). To achieve sufficient throughput in our analysis, we counted the number of vesicles being part of packets or residing in their proximity (designated vesicle

packet elements, VPEs) in a total of 40 cell profiles (Fig. 6d, f). In two independent experiments, obtained values were normalized to the transfection efficiency of each sample to allow comparison between different samples (Fig. 6f). For the WT polyprotein and the A137T polymorphism in NS4B, an almost complete loss of VPE formation was observed in JNJ-A07 treated cells, consistent with the compound sensitivity in transient replication assays[37]. Importantly, VPE efficiency was not affected for any of the NS4B resistance mutants by JNJ-A07 treatment using a concentration of 35 nM. In addition, VPEs of all samples were analyzed regarding their morphology (Fig. 6g). While JNJ-A07 had no effect on VPE diameter, some statistically significant deviations were found. However, there was no correlation between diameter alteration and the degree of compound resistance, suggesting that these two features are most likely not linked to the mechanism-of-action of JNJ-A07. Nevertheless, this result underscores that NS4B is involved in the formation of VPs.

## The interaction between DENV NS2B/NS3 and NS4A-2K-NS4B is functionally linked to the formation of vesicle packets

Next, we wondered whether the NS2B/NS3 - NS4A-2K-NS4B interaction is functionally coupled to the formation of DENV VPs. To this end, we used NS4B mutants we had reported earlier to abolish DENV RNA replication, but being rescued by second-site compensatory mutations[21]. The two primary mutations blocking viral replication were Q134A and M142A, both drastically reducing the interaction between NS2B/NS3 and NS4A-2K-NS4B[21]. Replication of Q134A is rescued by A69S and A137V, while replication of M142A is rescued, at least in part, by the JNJ-A07 resistance mutation L94F and by T215A, which confers resistance to NITD-688 (Fig. 7a, b)[41].

When tested in the trans-cleavage assay (Fig. 7c), the mutations Q134A and M142A in NS4B significantly reduced the interaction between NS4A-2K-NS4B and NS2B/NS3 (Fig. 7d, e). Combination of Q134A with the pseudo-reversions led to almost complete recovery of this interaction, while partial restoration was found for the M142A mutant when combined with the pseudo-reversion L94F, although it did not reach statistical significance (Fig. 7e). Nevertheless, this level of restoration correlates well with the replication competence of the mutants as reported earlier[21]. Notably, a striking correlation was found when these mutants were analyzed for their competence to form VPs (Fig. 8a–c). While no vesicles were found for the Q134A mutant, insertion of the two pseudo-reversions A69S and A137V restored VPE formation to WT level (Fig. 8d). The NS4B mutation M142A also reduced efficiency of VPE formation by more than 99%, which was restored to about 60% of the WT efficiency when the two pseudo-reversions were combined with this primary mutation (Fig. 8d). Interestingly, tubular membrane structures were very frequent in this triple mutant, while usually accounting for only a few percent in other samples (Fig. 8e, f). Whether RNA replication of this mutant occurs at these tubular structures, or the lowly abundant, regular VPs remains to be determined. Collectively, these data show that the interaction between the NS4A-2K-NS4B

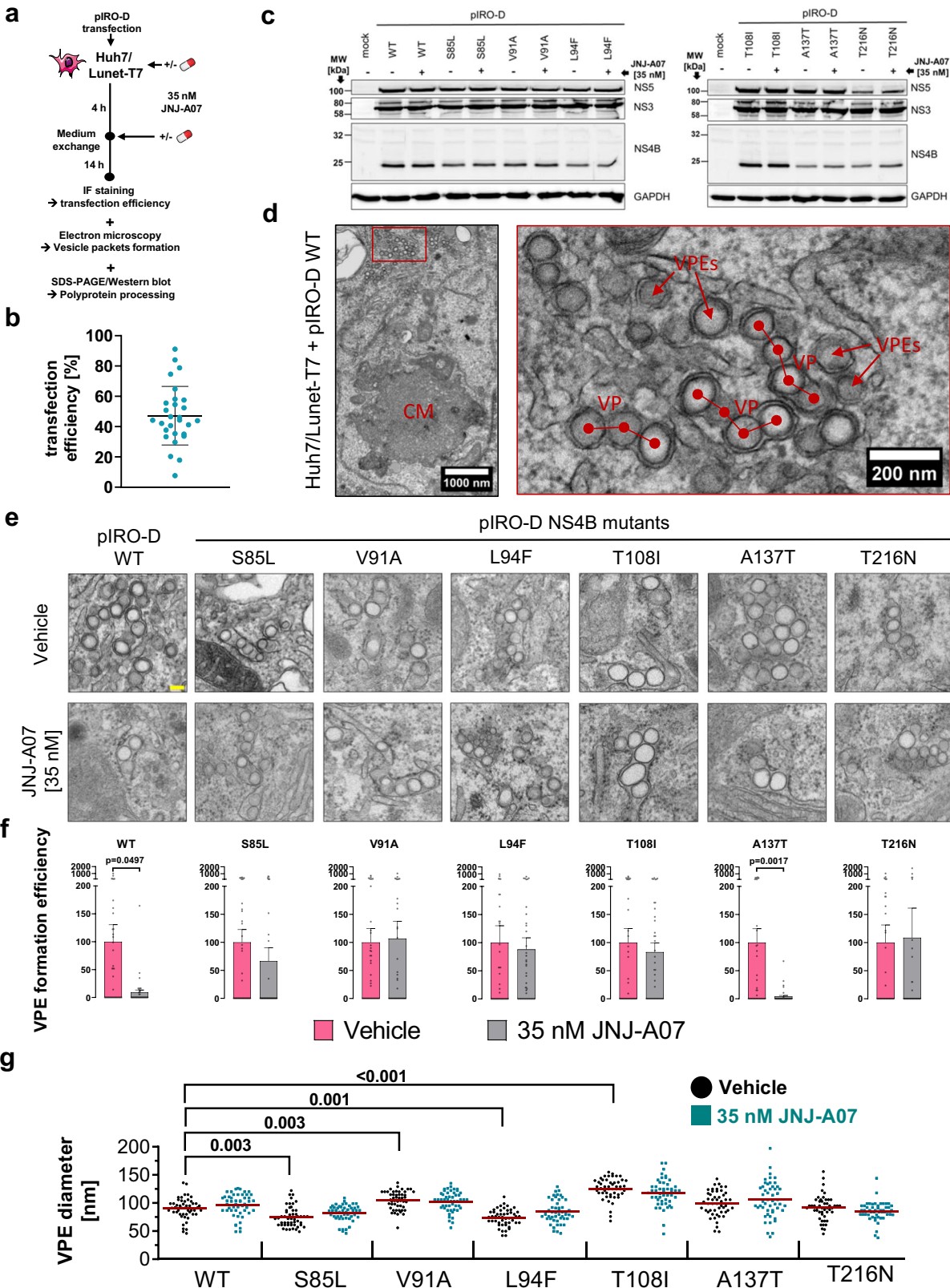

**precursor and the NS2B/NS3 complex is functionally coupled and causally related.**

## JNJ-A07 does not disrupt established DENV replication organelles

The in vitro antiviral effect of JNJ-A07 declines with increasing time between infection and compound addition to the cells[37]. Moreover, we

found that JNJ-A07 does not disrupt existing complexes of NS4A-2K-NS4B and NS2B/NS3, but rather blocks their de novo formation[37]. Since we found a link between the loss of this interaction and VPs formation, we wondered whether JNJ-A07 selectively inhibits de novo formation of vesicles rather than disrupting established ones. In the first set of experiments, we used TEM to compare cells transfected with the NS1-5 encoding pIRO-D plasmid and treated with JNJ-A07 4 h or 16 h after

**Fig. 6 | JNJ-A07 inhibits the formation of VPs in WT but not in compound-resistant NS4B mutants. a** Experimental workflow. Huh7/Lunet-T7 cells were transfected with pIRO-D plasmids encoding DENV NS1-NS5 WT or containing compound resistance mutations in NS4B. Cells were treated with JNJ-A07 or vehicle and 14 h later, fixed for immunofluorescence (IF) and transmission electron microscopy (TEM) analysis or lysed for western blot. **b** Transfection efficiencies as derived from IF analyses. Shown is a combined scatter plot of transfection efficiencies across all 14 specimens over the two independent experiments presented with mean and SD. **c** Western blot of DENV proteins; GAPDH served as loading control. **d** TEM analysis of cells transfected with the WT pIRO-D construct. Detected morphological alterations comprise convoluted membranes (CM) and vesicle packets (VP), which are string-like arrays of vesicles, here denoted as vesicle packet elements (VPEs). Right panel: enlargement of the red boxed area in the left panel. **e** Representative electron micrographs showing VPEs detected in cells after transfection with constructs specified on the top. Yellow scale bar = 100 nm. **f** Quantification of VPEs from $2 \times 20$ cell profiles per transfected cell sample ($n = 2$ independent experiments), after normalization to the transfection efficiency as determined by IF (see panel b). For each construct, VPE formation efficiency determined with vehicle-treated control cells was set to 100. Results are shown with mean and SEM. *P*-values were calculated by Dunn's multiple comparison test following a Kruskal-Wallis test. **g** For each sample, the diameter of 50 randomly selected VPEs was determined. Shown are the individual values with mean and standard deviation. A two-way ANOVA and subsequent Šidák's multiple comparisons test were employed for statistical analysis.

transfection (Fig. 9a). While in the 4 h-treatment setting, the compound is present at a time when VPs are forming, in cells treated 16 h after transfection, VPs have already formed prior to compound addition. In both conditions, polyprotein processing was not affected (Fig. 9b). We observed an almost complete loss of VPs in cells treated early after transfection, while in cells in which JNJ-A07 was added late after transfection, VPE formation efficiency was reduced to a much lower degree (Fig. 9c, d).

A limitation of this time-of-addition experiment was that cells were exposed to the compound for different time spans (16 vs. 4 h in the early vs. late condition, respectively; Fig. 9a). Therefore, firm conclusions about inhibition of de novo synthesis versus active disruption of VPs by JNJ-A07 could not be made. To overcome this confounding effect, we performed an additional experiment in which protein synthesis was blocked by cycloheximide (CHX) at a time point when VPs had already formed (Fig. 9e). Thus, de novo formation of additional vesicles was inhibited, allowing the selective analysis of their natural decay. A ribopuromycylation assay was used to ensure that the applied amount of CHX completely arrested RNA translation (Fig. 9f). TEM analysis showed that in cells treated with JNJ-A07 and CHX, the number of VPEs was comparable to those treated with CHX, arguing that JNJ-A07 does not accelerate VPE decay (Fig. 9g, h).

Taken together, these results suggest that the interaction between NS2B/NS3 and NS4B-containing proteins is required for VPs formation. We propose that JNJ-A07 blocks de novo VPs biogenesis, but does not disrupt existing ones, consistent with the prevention, but not disruption of the NS2B/NS3 interaction with NS4B by the compound.

### The mechanism-of-action of the NS4B inhibitor NITD-688 overlaps with that of JNJ-A07

Recently, another pan-serotype DENV inhibitor termed NITD-688 (Supplementary Fig. 1b) was described that targets NS4B and has a resistance profile partially overlapping with that of JNJ-A07[41]. Since no mechanism-of-action has been described for this compound, we investigated whether NITD-688 acts similarly to JNJ-A07. For comparative analysis, we first studied the possible impact of NITD-688 on the NS2B/NS3 interaction with NS4A-2K-NS4B in the trans-cleavage system (Supplementary Fig. 6a) using various compound concentrations (from 1 nM to 10,000 nM). The two JNJ-A07-mediated phenotypes in this system, i.e., loss of NS2B/NS3 co-precipitation with NS4A-2K-NS4B and precursor processing (as seen by a reduction of the 2K-NS4B signal) could be reproduced for NITD-688 (Supplementary Fig. 6b, c). However, much higher compound concentrations were required (EC$_{50}$ around 3,900 nM and 500 nM, respectively), consistent with the ~900-fold lower antiviral activity of NITD-688 compared to JNJ-A07[37]. Importantly, when analyzing the effect of NITD-688 on DENV VPs formation, we found that this compound also prevents the de novo biogenesis of VPs whereas established ones were affected to a much lesser extent (Supplementary Fig. 6d–g). In summary, these results suggest that the pan-serotype DENV inhibitors NITD-688 and JNJ-A07 share a similar mechanism-of-action by blocking the interaction between NS2B/NS3 and NS4A-2K-NS4B and preventing the formation of VPs, the putative sites of DENV RNA replication.

## Discussion

In this study, we characterized the molecular mechanism of JNJ-A07. Consistent with the assumption of targeting NS4B, we observed specific accumulation of JNJ-A07-related PAL compounds at intracellular sites enriched for NS4B and its interaction partner, the NS2B/NS3 complex. Whether these structures correspond to convoluted membranes, VPs or some other structures remains to be determined. Since NS4B most likely contributes to VPs formation, one would assume compound accumulation also there, yet VPs are too small to be visualized by light microscopy and labeling methods suitable for correlative imaging and retaining membrane integrity are not available. Notably, beyond the spatial proximity of inhibitor and NS4B-rich sites, we show compound binding to the NS4A-2K-NS4B precursor and mature NS4B arguing for a binding site present in both protein species. This finding is in line with NITD-688, for which compound binding to NS4B has been experimentally proven by means of NMR while binding to the precursor has not been studied[41]. We note that our results point to the interaction between the NS2B/NS3 complex and the NS4A-2K-NS4B precursor as being the critical target of JNJ-A07. Additional binding to mature NS4B does not contradict this, since binding alone does not imply pharmacological significance.

A stable interaction between the NS4A-2K-NS4B precursor and NS2B/NS3 has not yet been described in the literature. Only a short-lived enzyme-substrate interaction between the precursor and the NS3 protease domain is known to be required for cleavage at the NS4A-2K junction (first described for YFV[22] and later confirmed for other flaviviruses including DENV[21,23]). Indeed, both here and in previous publications, we observed an altered cleavage pattern of NS4B species in cells treated with JNJ-A07 when using the trans-cleavage system[37]. In response to our first publication on JNJ-A07, Behnam and Klein published an opinion article postulating that JNJ-A07 alters the canonical cleavage sequence of NS4A-2K-NS4B via 2K-NS4B to mature NS4B in such a way that the host signal peptidase complex directly cleaves off NS4B from NS4A-2K-NS4B giving rise to uncleaved NS4A-2K[44]. However, the results reported here reject this hypothesis. First, analysis of HA-NS4A-2K-NS4B precursor cleavage by NS2B/NS3 in the presence of JNJ-A07 did not reveal the postulated NS4A-2K protein, but only fully cleaved NS4A. Second, co-expression of the NS4A-2K-NS4B precursor with a catalytically inactive NS2B/NS3 protease in the presence of JNJ-A07 resulted in a dose-dependent loss of precursor–protease interaction, but not in the formation of mature NS4B that, according to Behnam and Klein[44], should be released independent from the viral protease by the host cell signalase. Third, the same result was found when stabilizing the NS4A-2K-NS4B precursor via a mutation affecting the NS4A-2K cleavage site. Finally, cleavage kinetics of the NS4A-2K-NS4B precursor in the polyprotein context (pIRO-D system) were not affected by JNJ-A07, as determined in the pulse-chase experiments. Taken together, our

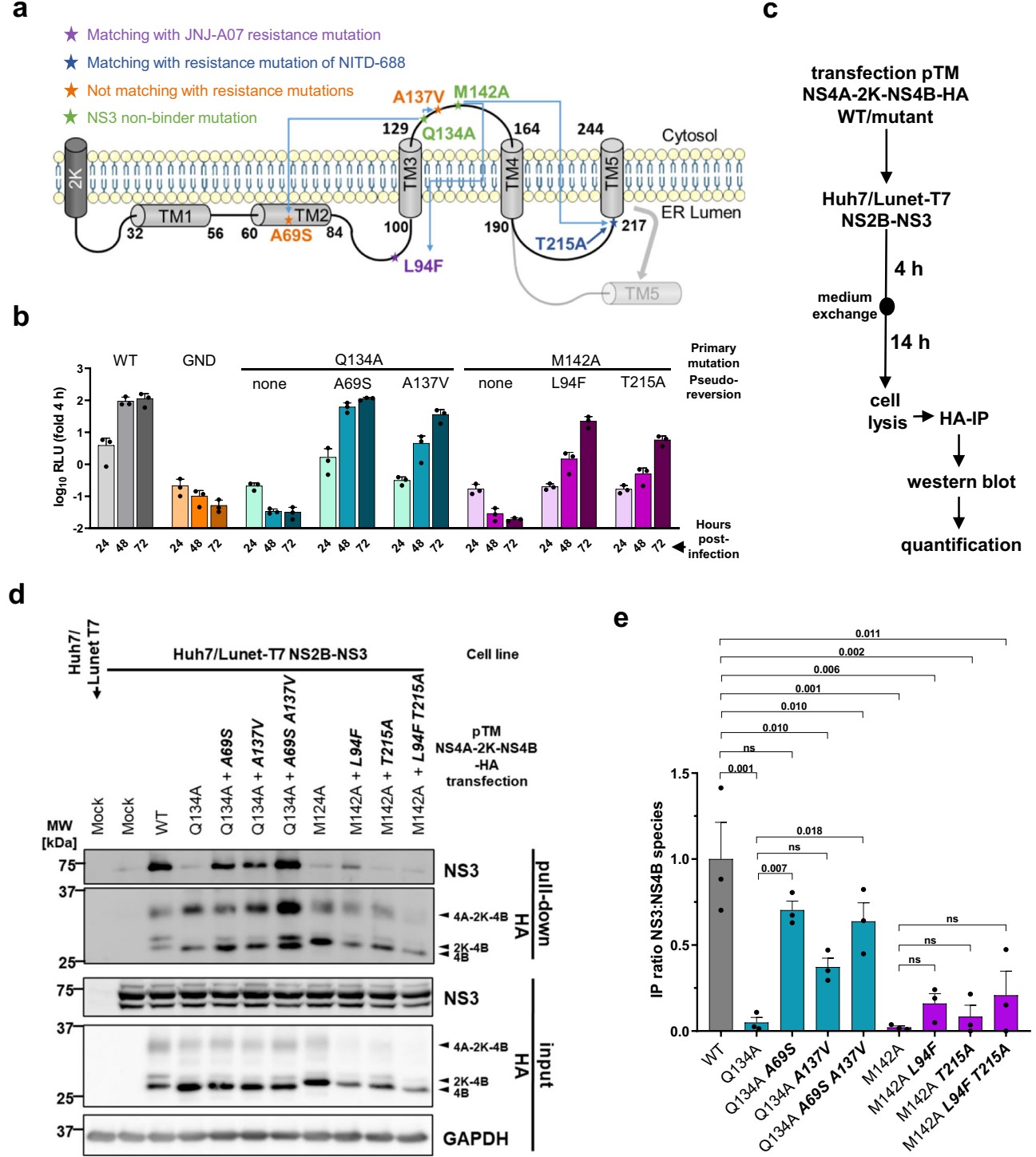

**Fig. 7 | Pseudoreversions in the NS3 non-binder mutants NS4B Q134A and M142A rescue the NS3–NS4B interaction. a** Membrane topology of NS4B according to Miller et al[43]. and position of NS3-non-binder mutations Q134A and M142A (in green), along with their corresponding pseudo-reversions. **b** Replication kinetics of sgDVs-R2A replicons harboring the NS4B mutations Q134A or M142A with and without second-site pseudo-reversions. Adapted from Chatel-Chaix et al[21]. WT and GND mutant served as positive and negative control, respectively. Luciferase values were normalized to the 4 h time-point reflecting transfection

efficiency. Data are *n* = 3 and are presented with mean and SD. **c** Experimental approach to study the capability of pseudo-reversions to restore the interaction between NS4A-2K-NS4B and NS2B/NS3. **d** Representative western blot (*n* = 3 independent experiments). HA-captured protein complexes were analyzed along with the corresponding input (20%). GAPDH served as loading control. MW, molecular weight. **e** IP ratio of NS3 to NS4B-containing species (mean and SEM) from 3 independent experiments. For statistical significance analysis, a repeated measures one-way ANOVA with subsequent Šidák's multiple comparisons test was applied.

results provide compelling evidence that JNJ-A07 does not exert its antiviral effect via an altered cleavage sequence.

Comparison of the impact of JNJ-A07 on the interaction between DENV proteins in the trans-cleavage and the polyprotein system

revealed that the known interaction between NS4A-2K-NS4B with NS1 was not affected in both systems, while the interaction of the precursor with NS2B/NS3 was prevented in the trans-cleavage system but appeared unaltered in the polyprotein system. Several possibilities

might account for this difference: In the trans-cleavage system, processing of the NS4A-2K-NS4B precursor appears to be rather inefficient as deduced from the high amounts of uncleaved precursor. In addition, NS4A-2K-NS4B is expressed independent of the NS2B/NS3

protease and therefore, this precursor might be expressed in a much higher molarity than the enzyme, causing additional accumulation of unprocessed substrate. In contrast, the cleavage kinetics in the polyprotein are very rapid, which is due to the fact that many processing

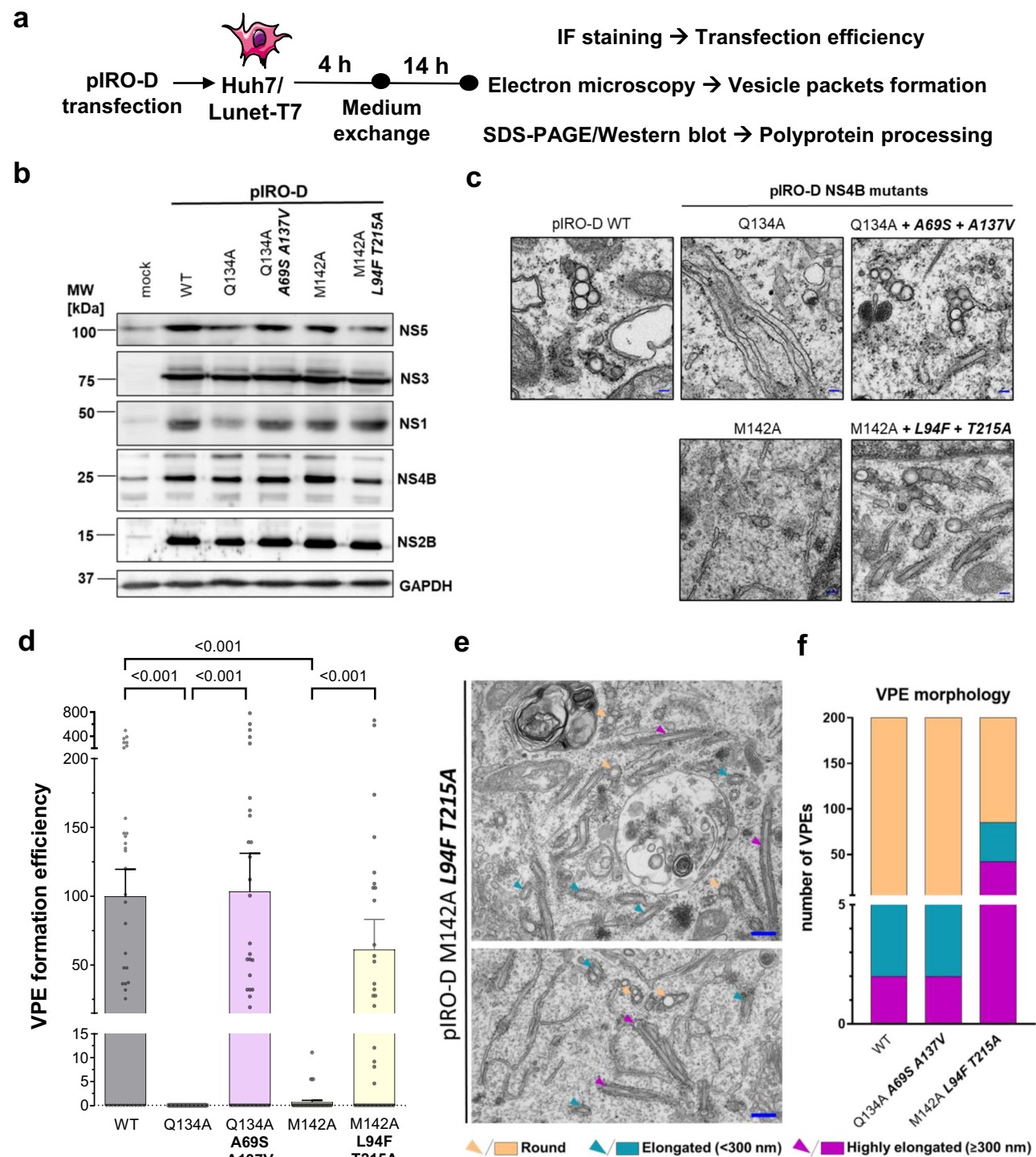

**Fig. 8 | Interaction between NS2B/NS3 and NS4A-2K-NS4B is functionally linked to vesicle packets formation. a** Experimental approach to analyze VPs formation by NS3 non-binder mutants and pseudo-revertants. Huh7/Lunet-T7 cells were transfected with pIRO-D plasmids encoding DENV NS1-NS5 WT or NS3 non-binder mutations in NS4B (with or without compensating pseudoreversions) and fixed 18 h later for immunofluorescence (IF) and transmission electron microscopy (TEM) analysis, or lysed for western blot. **b** Representative western blot (*n* = 2 independent experiments). **c** Representative electron micrographs. Scale bar =

100 nm **d** Quantification of VPEs (20 cell profiles for each experiment). Values were normalized for transfection efficiency and are plotted with mean and SEM. *P*-values were calculated using Dunn's multiple comparison test after a Kruskal-Wallis test. **e** Example images from two independent cell profiles illustrating vesicle elongation in pIRO-D NS4B M142A L94F T215A transfected cells. Colored arrowheads are specified on the bottom. Scale bar = 300 nm. **f** Quantitative comparison of VPE morphotypes in WT and triple mutant transfected cells. 200 VPEs per sample were analyzed. Compensatory pseudo-reversions are highlighted by bold and italic letters.

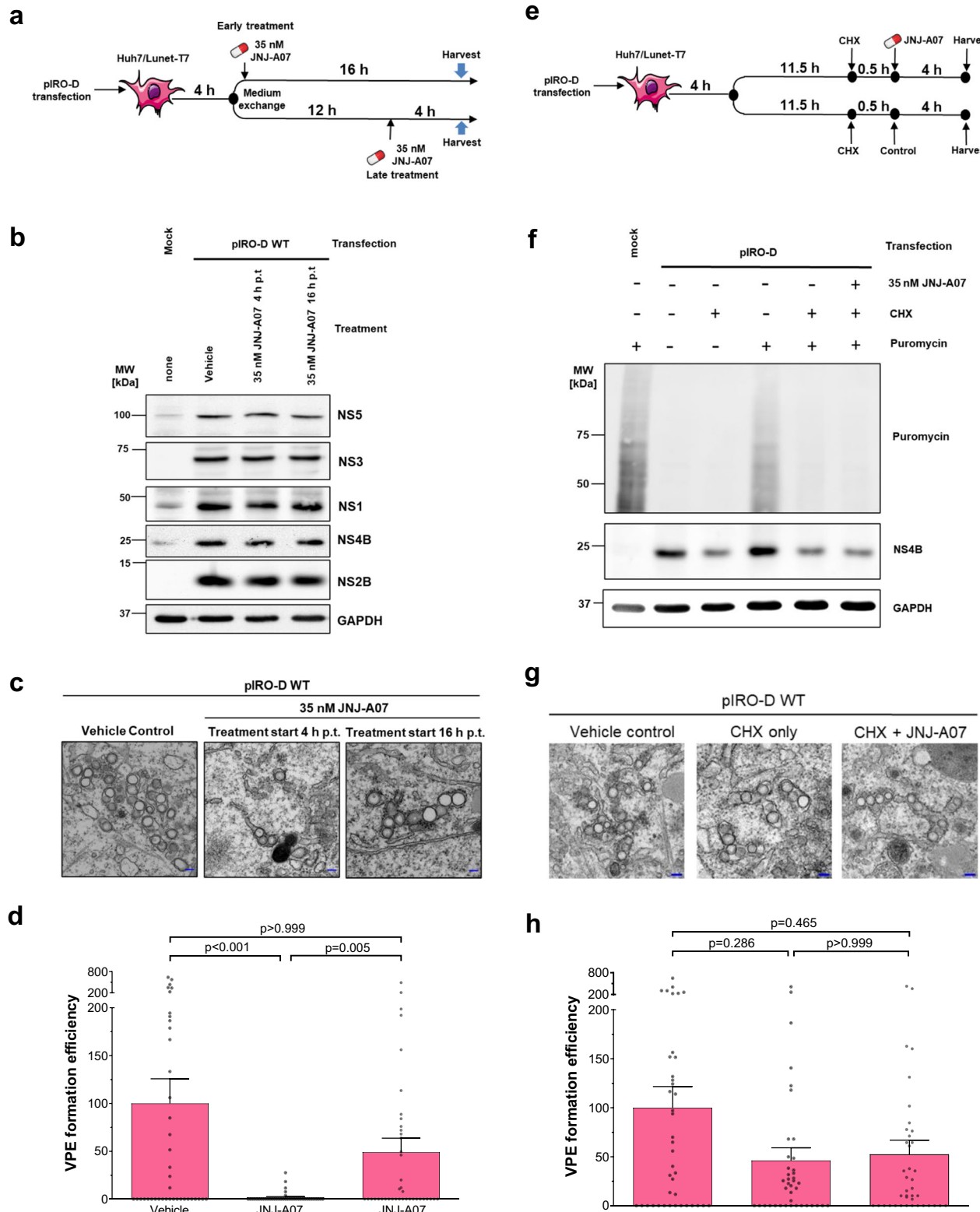

steps occur predominantly in cis as the protease is part of its own substrate[13,45]. Although JNJ-A07 affects the interaction of NS2B/NS3 with the NS4A-2K-NS4B precursor, but not with fully cleaved NS4B, compound-mediated effects on precursor processing could not be detected in the pIRO-D system, neither by western blot measuring the steady-state (Fig. 6c), nor by using more sensitive radiolabeling of nascent DENV proteins (Fig. 3).

Results obtained with replication deficient NS4B mutants (Q134A, M142A) and corresponding pseudo-revertants provide compelling evidence that interaction between NS2B/NS3 and NS4A-2K-NS4B plays a critical role in the formation of VPs, the sites of viral RNA replication. The functional link between the NS2B/NS3 interaction with the NS4A-2K-NS4B precursor on the one hand, and ER membrane rearrangement on the other hand is supported by the data obtained with the JNJ-A07

**Fig. 9 | JNJ-A07 blocks de novo formation of vesicle packet elements but does not disrupt existing ones. a** Experimental approach. Cells were harvested and analyzed by TEM, WB and IF to determine VPE formation, polyprotein expression and processing as well as transfection efficiency, respectively. **b** At the tested concentration of 35 nM, JNJ-A07 shows no apparent effect on polyprotein processing, regardless of the time of addition. Shown is one of two western blots. **c** Example electron micrographs. Scale bar = 100 nm. **d** VPE formation efficiency with mean and SEM for early and late timepoint of compound addition, relative to vehicle-treated control cells. VPEs from 20 cell profiles per condition and independent experiment ($n = 2$) were quantified and normalized to transfection efficiencies. *P*-values were calculated by Dunn's multiple comparison test that was conducted after a Kruskal-Wallis test. **e** To discriminate between block of VPE de novo formation and active VPE disruption, cells were pre-treated with cycloheximide (CHX; 200 μg/ml) 30 min prior to JNJ-A07 addition. Cells treated only with CHX served as reference. **f** Representative western blot of a ribopuromycilation assay showing CHX-mediated inhibition of protein synthesis. Cells were treated with puromycin (1 g/ml) 5 minutes before harvest. Translation rate is reflected by the amount of puromycin incorporated into polypeptide chains, which is detected by puromycin-specific western blot. **g** Exemplary electron micrographs showing VPEs in transfected cells as outlined in **e**. Scale bar = 100 nm. **h** VPE formation efficiency (mean and SEM). Values were normalized to vehicle-treated cells (vehicle) that were set to 100%. Data acquisition and statistical analysis were performed as for **d**.

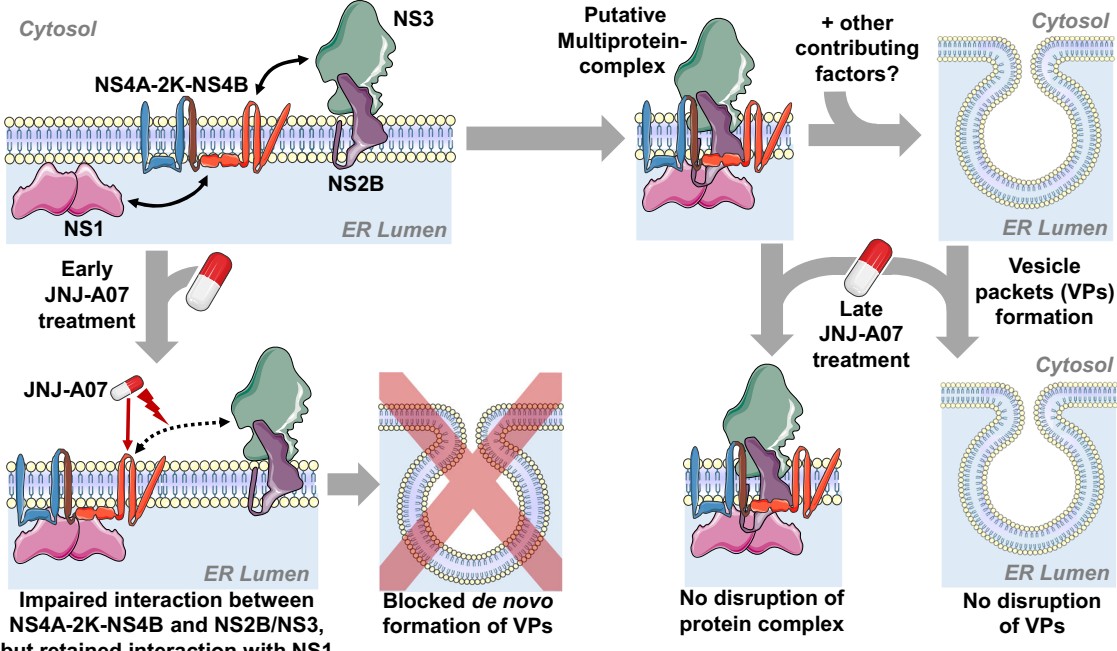

**Fig. 10 | Graphical summary of the mechanism-of-action of JNJ-A07.** Upper panels (left to right): During DENV polyprotein processing, the cleavage intermediate NS4A-2K-NS4B is formed, which engages interaction with NS1 and the viral protease complex NS2B/NS3. This hypothetical multiprotein complex, together with other factors (e.g. certain NTR regions of the viral genome and putative host cell factors), induced de novo formation of VPs, the assumed sites of viral RNA replication. Lower panels (left to right): When applied early, JNJ-A07 might bind specifically to solitary NS4A-2K-NS4B and impairs its interaction with NS2B/NS3, whereas binding of NS1 on the ER luminal side is not affected. This prevents de novo formation of VPs, thus blocking viral RNA replication. When JNJ-A07 is added after VPs have already formed, they remain intact and decay with their natural half-life. This assumption is consistent with our previous observation that JNJ-A07 cannot break established NS4A-2K-NS4B - NS2B/NS3 complexes and with the known gradual loss of antiviral activity of JNJ-A07 with increasing time between the onset of viral replication and compound application.

resistance mutants rendering both the above-mentioned interaction and VPs formation compound resistant. This link would also explain why the compound is only active at early time points after infection, i.e. by the time when VPs form. Later on, when the VPs are established, JNJ-A07 does not affect them and they decay with the natural turn-over rate, consistent with our observation that the compound prevents the NS2B/NS3 interaction with NS4A-2K-NS4B, but does not disrupt the complex once it has formed (Fig. 10).

While there is no other known association of the NS4A-2K-NS4B precursor with NS2B/NS3 reported in the literature, in vitro studies argue that mature NS4B acts as cofactor of the NS3 helicase to coordinate helicase ATP cycles and nucleic acid binding in the context of viral genome unwinding[28,31]. Interestingly, NS4B residues reported for helicase interaction correspond largely to those NS4B amino acid residues altered in NS3 non-binding mutants characterized by us in this and previous work[21]. Although we cannot exclude that NS3 ATPase/ helicase activities are required for VPs formation, we consider this unlikely because VPs can be induced in the absence of viral RNA replication. In any case, our study reveals a hitherto undescribed function of the NS4A-2K-NS4B precursor in VPs formation for which its interaction with the NS2B/NS3 complex is required.

The DENV inhibitors JNJ-A07 and NITD-688 share some resistance mutations residing close to each other at the beginning of transmembrane domain 5 of NS4B (NITD688: T215 A/S; JNJ-A07: T216N/P)[41]. Likewise, resistance-associated mutations against the YFV-specific inhibitor BDAA reside at the corresponding site of YFV NS4B (P219S)[46]. For the latter, a reduction of the number of observed VPs has been reported[40]. However, in these studies the effect was measured in infected cells and therefore, it remains to be determined whether reduced number of VPs is mainly an indirect phenotype resulting from suppressed viral replication or results from direct inhibition of VPs biogenesis or accelerated degradation. Consistent with the latter, it has been reported that in cells treated with BDAA an innate cytokine response is triggered via the sensing of double-stranded viral RNA through cellular pattern recognition receptors[40]. Taken together, this study and our data suggest that JNJ-A07, NITD-688, and BDAA all target certain NS4B-containing species and impair flaviviral replication organelle formation or integrity.

A holistic view of the mechanism-of-action of JNJ-A07 reveals striking parallels to hepatitis C virus (HCV) inhibitors targeting non-structural protein 5A (NS5A)[47] and representing a central pillar of direct-acting antivirals used to treat chronic hepatitis C patients. Like DENV NS4B (or the NS4A-2K-NS4B precursor), HCV NS5A is an atypical drug target as it lacks known enzymatic activity and is generated by regulated cleavage of an NS4B-NS5A precursor[48]. HCV NS5A binds multiple viral and host cell factors and exerts various functions depending on the interaction partner. Using an expression-based system, we have earlier shown that an analogue of the NS5A inhibitor daclatasvir prevents the formation of the presumed HCV replication organelle (designated membranous web), which was not the case with a drug-resistant NS5A mutant[49], similar to what we report here for JNJ-A07. In conclusion, these results suggest that non-enzymatic viral proteins acting as central organizers in viral replication are highly promising drug targets.

## Methods

### Compounds

The synthesis of JNJ-A07 is published in WO/2017/167951 (example 4B)[50]. The 3-Acyl-indole Compounds A and B are substituted at the indole-N position with a lipophilic tail that is functionalized with a diazirine moiety for photo-crosslinking and a terminal alkyne group for pull-down via copper-mediated click chemistry. Compounds A and B are respectively the more and less potent (R)-enantiomer and (S)-enantiomer and can be obtained by chiral separation from the racemic parent. A detailed description of the chemical synthesis of Compound A and B can be found in the Supplementary Methods. The synthesis of the DENV NS4B inhibitor NITD-688 was carried out at Janssen Pharmaceuticals, Inc. following a synthetic route as described in the literature[41]. For in vitro experiments, compounds were dissolved in 100% dimethylsulfoxide (DMSO) as a 5 mM stock.

### Cells

Human hepatoma (Huh7) and Huh7/Lunet cells were cultured at 37 °C and 5% $CO_2$ in DMEM, supplemented with 10% FBS, 2 mM L-glutamine, 100 U/mL penicillin, 100 µg/mL streptomycin and 1% nonessential amino acids (in further sections referred to as $DMEM_{Complete}$). Variants of these cell lines additionally expressing the bacteriophage T7 RNA polymerase along with a zeocin resistance gene have been described earlier and were passaged in the above mentioned medium complemented with 5 µg/mL zeocin[21]. Huh7-T7 NS2B-NS3, Huh7/Lunet-T7 NS2B-NS3, Huh7/Lunet-T7 NS1-HA and Huh7/Lunet-T7 Calnexin-HA cells were generated by lentiviral transduction as reported earlier[18,21] and were cultured in the additional presence of 1 µg/mL puromycin. Cell lines were regularly tested for mycoplasma contamination.

### Antibodies

Immunofluorescent staining of DENV proteins was performed using a mouse monoclonal DENV NS3 antibody (GTX629477, Clone GT2811, Lot# 42961, GeneTex, 1:200) and a rabbit polyclonal DENV NS4B antibody (GTX124250, Lot# 40779, GeneTex, 1:200) while ZIKV NS4B was stained with a rabbit polyclonal antibody (GTX133321, Lot# 42697, GeneTex, 1:200). The subsequent incubation with secondary antibodies was performed with either Goat anti-Mouse IgG (H + L) Alexa Fluor™ 488 (A-11029, Lot# 2120125, Invitrogen, 1:500) or Donkey anti-Rabbit IgG (H + L) Alexa Fluor™ 568 (A-10042, Lot# 2306809, Invitrogen, 1:500), depending on the species of the primary antibody. Mouse monoclonal anti-HA agarose beads, clone HA-7 (A2095, Lot# 119M4756V, Sigma-Aldrich), were utilized for immunoprecipitation experiments. In western blot staining, glyceraldehyde-3-phosphate dehydrogenase (GAPDH) or β-Actin served as a loading control for cell lysates (input) visualized with mouse monoclonal anti-GAPDH, G-9 (sc-365062, Lot # I2320, Santa Cruz Biotechnology, 1:1000 dilution) or β-Actin (A5541, clone AC-15, Lot # 0000090942, Sigma-Aldrich, 1:4000

dilution), respectively. HA-specific signals were detected with a purified mouse anti-HA.11 epitope tag, clone: 16B12 (lot #: B276381, Bio-Legend, 1:1000). DENV-2 proteins NS1 (1:1000), NS3 (1:2000), NS4B (1:1000) and NS5 (1:1000) were stained with in-house generated rabbit polyclonal antibodies[43], while NS2B was stained with a commercial rabbit polyclonal antibody (GTX124246, Lot #40751, GeneTex, 1:1000).

### Plasmids

Overlap extension PCR was employed to both insert point mutations into pIRO-D plasmids as well as to introduce HA-tags into pTM plasmids. Fused PCR fragments were treated with appropriate restriction enzymes in parallel to the vector plasmid and recombined using T4 ligase. Plasmid amplifications were performed in competent *E.coli* DH5α cells. After large-scale plasmid DNA preparation, the complete insert region was evaluated by nucleotide sequence analysis.

### DNA transfection

Cells were transfected 24 h after seeding using the liposomal reagent TransIT-LT1 (Mirus, Madison, WI, USA) according to the manufacturer's instructions. Cell culture medium was replaced both before and 4 h post-transfection.

### Determination of compound resistance

A panel of mutant subgenomic DENV reporter replicons (sgDVs-R2A) encoding either the WT NS4B or a JNJ-A07-associated resistance mutation was used to determine Compound A resistance. Details of the constructs have been described previously[37]. For RNA transfection, subconfluent cell monolayers were harvested, washed once with PBS and resuspended in cytomix buffer (120 mM KCl, 0.15 mM $CaCl_2$, 10 mM potassium phosphate buffer, 25 mM HEPES, pH 7.6, 2 mM EGTA, 5 mM $MgCl_2$, freshly supplemented with 2 mM ATP and 5 mM glutathione) at a density of $1 \times 10^7$ cells/ml. 10 µg of in vitro-transcribed RNA was mixed with 400 µl of cell suspension, transferred to an electroporation cuvette (Bio-Rad; 0.4-cm gap width), and pulsed once with a Gene Pulser II system (Bio-Rad) at 975 µF and 270 V. Immediately after pulsing, cells were transferred to prewarmed medium. Transfected cells (4000 cells per well in a 384-well plate) were incubated with serial dilutions of Compound A at 37 °C. 48 h post-transfection, viral replication was quantified by measuring luciferase activity using Renilla-Glo™ reagents (Promega) on a ViewLux™ Imaging System (PerkinElmer).

### Imaging based analysis of compound association with DENV proteins

Huh7/Lunet-T7 cells were seeded on coverslips in a 24-well format (35,000 cells/well). Transfection with pIRO-D plasmids (WT or compound resistance mutants) or pIRO-Z WT was performed as described above. Along with medium exchange (4 h post-transfection), cells were treated with either 1 µM Compound A, 1 µM Compound B or DMSO control (vehicle). 18 h after transfection, cells were briefly washed once with PBS and 1 ml of PBS was added to each well. The plates were then positioned without lids in a UV crosslinker (UVP crosslinker CL-1000M, Analytik Jena) and the cells were irradiated with light of wavelength 365 nm, transferring an energy of 200 $\frac{mJ}{cm^2}$ over a period of 5 min. Immediately thereafter, cells were fixed for 20 min at room temperature (RT) with a 4% paraformaldehyde (PFA) solution. Cells were permeabilized for up to 10 minutes at RT by incubation with a PBS solution containing 0.2% Triton X-100 before they were blocked for 1 h by treatment with 5% goat serum albumin (GSA) in PBS. After a brief wash in PBS, coverslips were placed upside down on 20 µl drops of click-reaction mixture (aqueous solution with 1 mM ascorbic acid, 1 mM $CuSO_4$·5 $H_2O$, 100 µM Tris(3-hydroxypropyltriazolylmethyl) amine (THPTA) and 50 µM Picolyl-Azide-Cy5.5 [Jena Bioscience]) in a humid, dark chamber. After one hour of incubation at RT, the reaction mix was removed, cells were washed and stained for NS3 and NS4B

using primary antibodies (1:200 in 5% GSA; 1 h). Following another wash, samples were treated with Alexa Fluorophore-conjugated secondary antibody dilutions (1:500 in 5% GSA; 30 min). Subsequently, the specimens were first washed 3 times with PBS, and then 3 times with water before being mounted onto microscope slides using DAPI-containing Fluoromount G. To determine subcellular colocalization between NS3, NS4B, and PAL compounds, 7 cells per sample and experimental run were imaged using a Leica SP8 confocal microscope. Pearson correlation coefficients were determined using the ImageJ plugin Coloc2 (ImageJ version 2.1.0/1.53j; Wayne Rasband and contributors, National Institutes of Health, USA). To categorize the morphological pattern of NS4B and to quantify the NS4B signal intensity, overview images were captured at 20× magnification on a Nikon Eclipse Ti microscope.

### Immunoprecipitation experiments

HA-immunoprecipitation experiments were carried out as previously described with some minor adjustments[37]. In brief, cells were exposed to transfection mix with indicated plasmids for 4 h before fresh DMEM$_{Complete}$ was added containing JNJ-A07 or an equivalent amount of DMSO as vehicle control. Cells were harvested 18 h post-transfection and lysed on ice for 20 min in 500 μL PBS-based lysis buffer containing 0.5% dodecyl beta-D maltoside (DDM; w/v) and protease inhibitors (Roche). To remove cell debris, lysates were centrifuged in a pre-cooled (4 °C) benchtop centrifuge for 45 min at maximum speed (21,130 × g). A Bradford Assay was used to determine the protein concentration of each sample and samples were adjusted to the one with the lowest concentration. A small fraction of the total lysate (20%) was saved for later input control on western blot. For HA-specific immunoprecipitation, 30 μL of equilibrated mouse monoclonal anti-HA agarose beads (Sigma-Aldrich) were added to each sample and incubated overnight at 7 °C on a rotating wheel. The beads were washed two times with lysis buffer, followed by two more washes with PBS, including a tube exchange. Captured proteins were eluted at 37 °C (10 min) in 120 μl of PBS containing 5% SDS followed by an additional elution step in 100 μl of pure PBS. Four sample volumes of acetone were added to the combined eluates to allow overnight precipitation of proteins at −80 °C. Samples were subjected to centrifugation at 21,130 × g for 1 h at 4 °C. The resulting pellets were air-dried and reconstituted in SDS sample buffer. Cell lysates and captured protein complexes were analyzed by western blot using the antibodies specified in each experiment. Bands intensities were quantified using ImageJ (version 2.1.0/1.53j; Wayne Rasband and coworkers, National Institutes of Health, USA). To evaluate NS4B processing, the individual intensities of NS4A-2K-NS4B, 2K-NS4B, and NS4B were quantified from one image and sum intensities were set to 100. To determine the ratios of co-precipitated NS3 to HA-specific signal, it was first ensured for both proteins that the different experimental runs contributed with comparable weights to the overall result. For this purpose, the NS3 signal of a given sample within an experiment was normalized to the cumulated NS3 signal of all samples in that experiment. The same procedure was applied to HA-tagged proteins. In a second step, protein ratios were calculated and normalized to the mean ratio of the respective control sample.

### Click labeling on agarose beads

Specified cells cultured in 10 cm diameter dishes were transfected with indicated plasmids. After 4 h, 1 μM Compound A or DMSO (as vehicle control) was added as part of a medium exchange. 18 h after transfection, the cells were washed with 5 mL PBS, and the open dishes were placed in a UV crosslinker (UVP Crosslinker CL-1000M, Analytik Jena). Cells were exposed to 365 nm UV light (200 $\frac{mJ}{cm^2}$) for 5 minutes before harvesting. The immunoprecipitation protocol described above was used up to the first washing of the beads with PBS. Next, PBS was replaced by 500 μl of a PBS-based solution containing 1 mM ascorbic acid, 1 mM CuSO$_4$·5 H$_2$O, 100 μM THPTA and 25 μM Picolyl-Azide-Cy5.5. In the absence of light, samples were incubated with the reaction mix for 3 hours at 7 °C on a rotating wheel, followed by 5 intensive washes with PBS including a tube change before SDS elution was performed as described above. After SDS-PAGE, in-gel fluorescence of the samples was recorded using a LI-COR Odyssey ® Imager (at 700 nm, close to emission peak of Cy5.5). Thereafter, a western blot transfer was performed and membranes were stained with an α-HA antibody to visualize NS4B-HA and NS4A-2K-NS4B-HA respectively.

### Determination of VPs formation

This procedure has been reported earlier[51]. In brief, Huh7/Lunet-T7 cells were seeded in 24-well plates and transfected as described above. Compound treatment with JNJ-A07, NITD-688, CHX or vehicle as well as sample collection was performed at different time points and is given in the schematic workflows on the respective figures. Transfection efficiency was determined using IF staining for NS3. Mock-transfected cells were used to define the background. Microscopic image acquisition was performed with a Nikon Eclipse Ti microscope in combination with a 20× air-objective. At least 1000 cells per condition were evaluated using a custom ImageJ macro (ImageJ version 2.1.0/1.53j; Wayne Rasband and contributors, National Institutes of Health, USA). To assess polyprotein processing, cells were washed with PBS, lysed in 100 μl 1x SDS-sample buffer containing benzonase (1 μl/well) and 20 μl per sample were used for SDS-PAGE. After transfer, membranes were cut and stained as indicated in the representative images. Cells prepared for EM analysis were washed with PBS and treated with fixative solution (50 mM cacodylic acid sodium trihydrate, 50 mM KCl, 2.6 mM MgCl$_2$, 2.6 CaCl$_2$, 1% paraformaldehyde [w/v], 2.5% glutaraldehyde [v/v]; 2% sucrose [w/v], 30 min at RT, then at 4 °C). Contrasting of the specimens was performed in two consecutive steps: first, with 2% osmium tetroxide in 50 mM cacodylate buffer (at least 40 min on ice); second, with 0.5% aqueous uranyl acetate (up to 16 h on ice). After removal of excess uranyl acetate with water, dehydration was carried out by serial exchange of the supernatant with ethanol solutions of increasing percentages (up to pure ethanol). Subsequently, cells were embedded in epoxy resin that was polymerized at 60 °C for at least 48 h. 70 nm thick sections were cut, transferred onto a formvar-coated TEM grid and post-contrasted with 3% aqueous uranyl acetate followed by saturated aqueous lead citrate for 5 min each. Air-dried grids were analyzed on a JEOL JEM-1400 transmission electron microscope. In each of the two experimental runs, 20 cell profiles were screened for VPEs, and the number of quantified vesicles was normalized to the respective transfection efficiency of the specimen. Vesicle diameters were determined with ImageJ.

### Ribopuromycylation assay

A protocol for the ribopuromycylation method to detect nascent protein synthesis[52] was integrated into the workflow for the determination of VPs formation. In brief, cycloheximide (CHX, 200 μg/ml) was added to cells 30 min before the start of a four-hour treatment with 35 nM JNJ-A07 (or vehicle control) to block de novo protein synthesis. Thereafter, the medium was replaced with DMEM$_{Complete}$ prewarmed to 37 °C, to which puromycin was added at a final concentration of 10 μg/ml. Cells were incubated at 37 °C for 5 minutes and then either fixed for microscopy or lysed for western blot analysis.

### Pulse-chase labeling

Huh7/Lunet-T7 cells were seeded into 6 well-plates (2 × 10$^5$ cells/well) 24 h prior to transfection. The transfection mixture was removed after 4 h by medium exchange, wherein the cells were treated with 35 nM JNJ-A07 or DMSO as vehicle control. 13 h later, cells were cultured in starvation medium for a further hour (DMEM lacking FCS, cysteine and methionine, but supplemented with 2 mM L-glutamine, 100 U/mL penicillin and 100 μg/mL streptomycin). After exchange to starvation

medium containing [$^{35}$S]-label mix (100 µCi/well), cells were incubated for 20 minutes (pulse). Immediately thereafter, cells were washed once with DMEM$_{Complete}$ and maintained in this medium until harvest at different time points (chase). In the case of JNJ-A07 treatment, the compound was re-added at 35 nM concentration during the starvation, labeling and incubation phases. Samples were analyzed using the standard protocol for immunoprecipitation except that no adjustment of protein concentration was performed and no control sample of total lysates was collected. After immunoprecipitation, 20 µl of each sample was subjected to SDS-PAGE. Gels were fixed in aqueous solution with 5% methanol and 10% acetic acid, vacuum dried onto filter paper and protein marker was labeled with radioactive ink (10 µCi/ml). Gels were analyzed by phosphoimaging using exposure times ranging from 3 to 20 days. Digital images were generated with a Bio-Rad Personal Molecular Imager FX system. The intensities of the NS4B-specific bands were quantified using ImageJ and adjusted to the number of cysteine and methionine residues contained in each protein (21 for NS4A-2K-NS4B, 14 for 2K-NS4B, and 13 for NS4B). For each time point, the percentage of NS4A-2K-NS4B precursor in the total amount of NS4B-specific species was calculated and its half-life time was determined by fitting "one-phase exponential decay" curves to the data sets using the GraphPad Prism 8.4.3 software package.

### Statistical analysis and data representation

Statistical analyses were performed using GraphPad Prism version 8.4.3. The tests used and corrections for multiple testing are specified in the legends of the respective figures. The same program was also used to create the scientific graphs of this work. *P*-values of ≤0.05 were considered significant.

### Reporting summary

Further information on research design is available in the Nature Portfolio Reporting Summary linked to this article.

## Data availability

All data supporting the findings of this study are available within the article. The source data file contains the raw data on which the graphs and diagrams are based, uncropped versions of any gels and blots shown in the figures as well as the replicates of analyzed gels and blots not shown in the figures. Source data are provided with this paper.

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

## Acknowledgements

This work was supported by a Seeding Drug Discovery Strategic Award from the Wellcome Trust (grant 089328/Z/09 and grant 106327/Z/14) and by the Flanders Agency Innovation & Entrepreneurship (VLAIO O&O grant IWT 150863). Work of R.B. was supported in part by the Deutsche Forschungsgemeinschaft (DFG, German Research Foundation) – project number 240245660 – SFB 1129 and project number 499982526 - BA 1505/11-1. For the publication fee we acknowledge financial support by Heidelberg University. We thank Marie Bartenschlager and Ulrike Herian from our research group for excellent technical assistance. We are grateful to Vibor Laketa for expert assistance and access to the Infectious Diseases Imaging Facility (IDIP) and to Charlotta Funaya for continuous support and access to the Electron Microscopy Core Facility (EMCF) at Heidelberg University. We thank Anne-Theres Henze, Benoit De Boeck and Bart Kesteleyn from Janssen Pharmaceutica for their competent help in specialist and administrative matters. In some figures, basic templates obtained from the Servier Medical Art library (https://smart.servier.com/) were used.

## Author contributions

Conceptualization, D.K., O.G., and R.B.; investigation, D.K., AS.H.K., S.D., U.H. and P.G.; provision of materials, JF.B., JI.A., B.G., M.vL. O.G. and R.B.; writing—original draft, D.K. and R.B.; writing—review and editing, D.K. AS.H.K., S.D., U.H., JF.B., JI.A., B.G., P.G., S.J.F.K., L.W., P.S., J.N., M.vL., O.G. and R.B.; funding acquisition, O.G., M.vL., and R.B.

## Funding

## Competing interests

J.-F.B., S.J.F.K., J.N., M.vL., and O.G. have been named inventor in a patent application claiming the discovery of this class of antiviral molecules as DENV replication inhibitors (WO 2017/167951), which was filed by applicants Janssen Pharmaceuticals, Inc. and Katholieke Universiteit Leuven, and has been granted in certain countries. O.G., J.-F.B., M.vL., S.J.F.K. and J.N. have been named inventors in a pending patent application relating to the use of substituted indole derivatives and substituted indoline derivatives in the manufacture of a medicament for the treatment or the prevention of dengue disease in an individual at risk of being infected by DENV and to a method for the treatment or the prevention of dengue in an individual at risk of being infected by DENV, which was filed by Applicants Janssen Pharmaceuticals, Inc. and Katholieke Universiteit Leuven (WO 2021/094563). O.G., B.G. and M.vL. are all full-time employees of Janssen and potential stockholders of Johnson and Johnson. The other authors declare no competing interests.

## Additional information

[1]Heidelberg University, Medical Faculty Heidelberg, Department of Infectious Diseases, Molecular Virology, Center for Integrative Infectious Disease Research, Heidelberg, Germany. [2]Janssen Infectious Diseases Discovery, Janssen-Cilag, Val de Reuil, France. [3]Evotec, Toulouse, France. [4]Discovery Chemistry, Janssen R&D, a Johnson & Johnson company, Toledo, Spain. [5]Discovery Chemistry, Janssen R&D, a Johnson & Johnson company, Spring House, PA, USA. [6]Discovery, Charles River Beerse, Beerse, Belgium. [7]Department of Microbiology, Immunology and Transplantation, Rega Institute for Medical Research, Laboratory of Virology and Chemotherapy, KU Leuven, Leuven, Belgium. [8]Leibniz Institute of Virology, Hamburg, Germany. [9]Janssen Global Public Health, Janssen Pharmaceutica NV, a Johnson & Johnson company, Beerse, Belgium. [10]German Centre for Infection Research, Heidelberg partner site, Heidelberg, Germany. ✉e-mail: ralf.bartenschlager@med.uni-heidelberg.de

