## [Peer Review File · Nature Communications]

Editorial Note: Parts of this peer review file have been redacted as indicated to avoid any copy right infringement.REVIEWER COMMENTS

Reviewer #1 (Remarks to the Author):

This manuscript intended to report the mode of action of the dengue virus inhibitor JNJ-A07. In the two Nature papers describing JNJ-A07 and its analog JNJ-1802, the concept that these compounds act through targeting the interaction between the NS3 and NS4B and preventing the formation of viral replication complex has been proposed. The current studies are natural extension of those published works, which provided more insightful molecular details in the mode of action of JNJ-A07. The study also supported the hypothesis that another clinical stage drug candidate NITD-688 may adopt a similar mode of action. The studies are comprehensively designed and performed with high scientific rigor.

The part of work to demonstrate that the interaction between NS3 and NS4B is functionally linked to de novo formation of vesicle packets (VPs) and that JNJ-A07 blocks VPs

biogenesis but not the established ones are solid and convincing.

The major weakness is in understanding the molecular detail of how JNJ-A07 disrupts the interaction between NS3 and NS4B containing protein precursors, and whether such disruption is fully responsible for its antiviral effect. The discrepancy in results obtained from the trans-cleavage system and polyprotein system (Supporting Figure 4c) should be more reasonably interpreted. Concerns are raised that the proposed interaction between NS3 and NS4A-2K-NS4B precursor could be due to an artifact of slower processing in the trans-cleavage system.

The click chemistry and colocalization data are novel and nice to have, but only provided indirect evidence of possible binding, since JNJ-A07's binding to other protein in the protein complex could not be formally excluded. Directly answering some of the related questions may improve the manuscript: Where does the compound bind? Does JNJ-A07 binding disrupt the interface/interaction between NS4B containing protein and NS3, which only transiently exists during replication organelle formation? A more in-depth discussion would be also helpful.

Minor:

Does compound A share the same resistance mutation profile with JNJ-A07? This is important in order to properly interpret the data in Figure 1. The difference in reduced colocalization did not correspond to the level of resistance conferred by the respective mutation to JNJ-A07.

Supporting Figure 2b. Label on the top of the panel should not be "WT".

Reviewer #2 (Remarks to the Author):

In this manuscript, the authors explore the mechanisms of how JNJ-A07 inhibits dengue virus infection. While most of the experiments had the appropriate controls, there are some important questions that should be addressed:

1. Are the drug assays performed at concentrations that are used physiologically, and what is the extent of virus inhibition for the drugs used in the experiments?
2. There are no experiments to directly prove that the drug impacts VP or VPE biogenesis. To prove this, the authors will need to label the VP before virus infection and visualise the labelled VP after infection to prove the role of drug in inhibiting VP formation.
3. If the drug truly inhibits VP biogenesis, wouldn't the cell undergo cell death under drug treatment? Did the investigators test for the health and viability of cells under drug treatment?
4. Where are the VPs localised? Are they in the ER, or with mitochondria that are known to interact directly with NS4B? The images taken with TEM should be able to shed insights on this.
5. Did the investigators see dengue virus colocalise with VP in drug and control conditions?
6. Using huh-7 and huh-7 subclone does not prove that the experiment is done on 2 cell types. They are essentially same type of cells, unless the investigators can prove that the 2 cells are genetically very different
7. If JNJ-A07 inhibits in a similar mechanism as NITD-688, why is it more superior in terms of safety and efficacy profile? Also, are the mechanisms of inhibition similar because JNJ-A07 and NITD-688 share similar structures?
8. Note that many other viruses require VP for efficient replication and infection. Does this mean JNJ-A07 would be effective against these viruses as well? The authors should elaborate or at least comment on these in the discussion section

Minor comments:

1. Figure 1f. The error bars look overlapping with each other yet the p-values are highly significant. Perhaps the individual points should be plotted to better understand why the comparisons show significance.

Reviewer #3 (Remarks to the Author):

The manuscript from Bartenschlager and colleagues addresses the mechanism by which a known inhibitor, JNJ-A07, affects dengue virus infection. The results complement and expand earlier studies by the group. The data clearly show that the inhibitor interferes with interaction between NS2B/NS3 complex and the intermediate cleavage precursor NS4A-2K-NS4B. The results provide additional insight into understanding processing of the DENV nonstructural polyprotein. Furthermore, the results provide in addition insight into the mechanism of an already identified inhibitor which has the potential to contribute to design of new effective inhibitors.

Comment

1. The data are strong, and experiments were carefully executed but the manuscript overall is very complex with the large amount of supporting data across eight large figures. The need to switch back and forth to the supplemental data is distracting. If only key, necessary selected pieces of data from the supplemental results were incorporated to the extent possible into the primary figures and/or reduction of the large amount of supplemental data would help.

Rebuttal letter to the reviewers

Reviewer #1 (Remarks to the Author)

This manuscript intended to report the mode of action of the dengue virus inhibitor JNJ-A07. In the two Nature papers describing JNJ-A07 and its analog JNJ-1802, the concept that these compounds act through targeting the interaction between the NS3 and NS4B and preventing the formation of viral replication complex has been proposed. The current studies are natural extension of those published works, which provided more insightful molecular details in the mode of action of JNJ-A07. The study also supported the hypothesis that another clinical stage drug candidate NITD-688 may adopt a similar mode of action. The studies are comprehensively designed and performed with high scientific rigor. The part of work to demonstrate that the interaction between NS3 and NS4B is functionally linked to de novo formation of vesicle packets (VPs) and that JNJ-A07 blocks VPs biogenesis but not the established ones are solid and convincing.

The major weakness is in understanding the molecular detail of how JNJ-A07 disrupts the interaction between NS3 and NS4B containing protein precursors, and whether such disruption is fully responsible for its antiviral effect. The discrepancy in results obtained from the trans-cleavage system and polyprotein system (Supporting Figure 4c) should be more reasonably interpreted. Concerns are raised that the proposed interaction between NS3 and NS4A-2K-NS4B precursor could be due to an artifact of slower processing in the trans-cleavage system.

The pIRO-D system is the most authentic expression-based system available for dengue virus, as demonstrated by its ability to induce the formation of authentic replication organelles. Hence, much of the experimentation in this study was conducted with this system.

Nevertheless, the trans-cleavage system is a useful and important system for mode-of-action analyses, since the reduced nature of this system allows more in-depth examination of selected molecular interactions of viral proteins. We agree with the reviewer's concern that the results of the trans-cleavage system cannot be generalized to the processes occurring in infected cells. Moreover, we agree that the composition of NS4B-containing species in the steady state of the trans-cleavage system argues for slower processing kinetics compared with the pIRO-D system, the latter having polyprotein cleavage kinetics comparable to those in infected cells. Nevertheless, we are convinced to have gathered several lines of evidence that our findings from the trans-cleavage system do fit well to the findings from the pIRO-D system. As now clearly stated in the revised manuscript, these arguments are:

- Impairments in the interaction between NS2B/NS3 and NS4A-2K-NS4B observed in the trans-cleavage system correlate perfectly well with impairments in vesicle packets formation detected with the pIRO-D system.
This is true for both JNJ-A07 and the NS3-nonbinding mutants NS4B Q134A and NS4B M142A. For the latter, we can also state that all phenotypes are linked to the cytosolic loop between NS4B TMD 3 and TMD 4. Thus, there is no condition known to us that leads to a loss of NS3 co-precipitation with NS4A-2K-NS4B in the trans-cleavage system without affecting vesicle packets formation in the pIRO-D system.
- JNJ-A07 resistance mutations as well as pseudoreversions in NS3-nonbinding mutants restored both vesicle packets formation and interaction between NS4A-2K-NS4B and NS2B/NS3.
- Compensatory pseudoreversions (L94F and T215A) in the NS3-nonbinding mutant M142A partially overlap with JNJ-A07- and NITD-688-associated resistance mutations.

- Both in the trans-cleavage and the pIRO-D system we did not detect a JNJ-A07-related impairment of the interaction between NS4A-2K-NS4B and NS1, arguing that compound treatment does not induce global misfolding of NS4A-2K-NS4B, but rather interferes with the NS3-interaction interface. Along these lines, we did not detect a significant impact of JNJ-A07 treatment on NS4A-2K-NS4B precursor processing in the authentic pIRO-D system, nor did we find an impairment of NS4B stability there as shown by our pulse-chase experiments.
- Although we could not demonstrate a blockade of the interaction between NS2B/NS3 and NS4A-2K-NS4B in the pIRO-D system, by means of pulse-chase kinetics we were able to show that NS4A-2K-NS4B is an extremely short-lived species in the pIRO-D system, explaining why the amounts required for the detection of this interaction cannot be reached with that system.

For all these reasons, we are convinced that our observations made with the trans-cleavage system do not represent artefacts and that we have exhausted experimental possibilities to exclude them.

The click chemistry and colocalization data are novel and nice to have, but only provided indirect evidence of possible binding, since JNJ-A07's binding to other protein in the protein complex could not be formally excluded. Directly answering some of the related questions may improve the manuscript: Where does the compound bind? Does JNJ-A07 binding disrupt the interface/interaction between NS4B containing protein and NS3, which only transiently exists during replication organelle formation? A more in-depth discussion would be also helpful.

We share the reviewer's view that evidence of compound binding is an important component of a mechanism-of-action analysis. Thus, we have invested great efforts in this direction utilizing Compound A for photoaffinity labeling. In the first submission draft of this manuscript we had already successfully shown that Compound A, a compound from the same chemical space as JNJ-A07 and with high structural similarity to it, spatially associates with DENV NS4B.

Demonstration of compound binding to NS4B and its precursor using a combined SDS-PAGE/Western blot readout turned out to be technically challenging. For reasons not fully understood, DENV NS4B, but especially the NS4A-2K-NS4B precursor, is prone to aggregation under normal click chemistry conditions (incubation of click reagents with sample lysates at room temperature).

To overcome this constraint, we pursued two parallel approaches:

- Approach A involved the UV cross-linking of Compound A to target structures and the subsequent (established) HA pulldown of NS4B and NS4A-2K-NS4B, followed by mass spectrometric (MS) analysis (see workplan in Rebuttal figure 1). The goal was to detect peptides that carry the additional mass of Compound A. We teamed up with Dr. Pietro Scaturro at the LIV in Hamburg. He is an expert in proteomics and has strong experience in flavivirus biology. Initially, a great deal of effort was put into obtaining a maximum number of tryptic peptides detectable by MS, which is a major challenge given the very hydrophobic property of NS4B. Once the detection was established, the sample set was measured. It turned out that the identification of a Compound A-associated peak was not possible, probably because Compound A might alter the retention time or charge state of the peptide to which it is bound, thereby preventing its detection by MS analysis. Thus, with our current MS pipeline, we are unable to identify compound binding to NS4B or its precursor.

Rebuttal Figure 1: MS approach to compound binding site identification.

a, Workflow for the detection of NS4B peptides bound to Compound A by MS. After UV crosslinking, HA-IP and optional SDS-PAGE, targeted proteomics was done using cells treated with one of the PAL enantiomers or DMSO control (vehicle). **b**, Comparison of antiviral activity and cellular cytotoxicity (CC) of Compound A and Compound B. Compound A represents the (R)-enantiomer and is approximately 70-fold more active than the (S)-enantiomer Compound B. **c**, Simplified schematic and hypothetical model of how PAL compounds (here simplified as blue/white cartoon drug) are covalently linked to amino acid moieties of NS4B. Upon UV irradiation, the diazirine group of the PAL compounds is converted into a reactive carbene that smoothly inserts into C-H, N-H, or O-H bonds of amino acid residues in single-digit Å proximity. The additional mass of the NS4B peptide(s) to which the compound is cross-linked should be detectable via mass spectrometry. Note that some peptides might not be traced due to unfavorable charge properties (indicated by dashed line).

- Approach B pursued the strategy to detect NS4B-bound Compound A via imaging. From the reported colocalization experiments, we knew that both a click reaction and NS4B staining are possible when proteins are prevented from aggregating by PFA fixation. Since this approach was not directly applicable to SDS-PAGE/Western blot readout, we conducted the click reaction, which in the original workflow was performed in the lysates before the HA-IP, with captured HA-NS4B bound to the beads. Although some degree of aggregation was still observed with the NS4A-2K-NS4B precursor, we were able to detect specific binding of the compound to both NS4B and its precursor, but not to the colocalizing ER membrane protein calnexin that was used as specificity control. We have integrated these results in the new main figure 2 and added an additional paragraph in the results section of the manuscript. We have also revised the discussion to address the additional points raised by the reviewer in this context.

Minor:

Does compound A share the same resistance mutation profile with JNJ-A07? This is important in order

to properly interpret the data in Figure 1. The difference in reduced colocalization did not correspond to the level of resistance conferred by the respective mutation to NJN-A07.

We thank the reviewer for raising this interesting point. JNJ-A07 resistance mutations were obtained using a lengthy *in vitro* resistance selection assay over the course of almost a year, underscoring that the compound has a high resistance barrier. A determination of resistance mutations induced by Compound A would therefore have exceeded the time frame of the revision, but given the high structural similarity, we hypothesize that we would have obtained the same or a very similar set of resistance mutations. Instead, we used a subgenomic replicon to determine the degree of resistance conferred by the same mutations against Compound A. The results are shown in a comparative table in Supporting Figure 2 of the revised manuscript (also shown below as Rebuttal Table 1). Although there were some differences, overall, the distinction of strong (L94F, V91A, T216N) and weak (S85L, T108I and A137T) resistance mutations was comparable for both compounds. However, the level of resistance did not correlate strongly with the colocalization of NS4B and Compound A. We conclude that within the scope of this study we cannot say whether reduced compound binding is a relevant resistance mechanism and will investigate this question in more detail in a follow-up study. Nevertheless, the newly obtained data on the resistance profile of Compound A point to a structure-resistance relationship.

[redacted]

Supporting Figure 2b. Label on the top of the panel should not be “WT”.

This is indeed a typo and has been corrected.

Reviewer #2 (Remarks to the Author)

In this manuscript, the authors explore the mechanisms of how JNJ-A07 inhibits dengue virus infection. While most of the experiments had the appropriate controls, there are some important questions that should be addressed:

1. Are the drug assays performed at concentrations that are used physiologically, and what is the extent of virus inhibition for the drugs used in the experiments?

In a previous publication by Kaptein et al. (Nature, 2021), we performed extensive studies on the antiviral activity of JNJ-A07 (here presented as Rebuttal Table 2). In Vero E6 cells, antiviral activity was determined against a panel of 21 clinical isolates covering the known natural diversity of DENV geno- and serotypes. In addition, we tested the antiviral activity of JNJ-A07 against the DENV-2 strain 16681 in several other cell types, including Huh7 cells used in this work. Therefore, we have consistently reported the compound concentrations used in multiples of the previously determined EC₅₀. The most common JNJ-A07 concentration used here was 35 nM, corresponding to about 45-fold EC₅₀.

[redacted]

For comparison, a dose typically used in mouse animal studies was 10 mg/kg, corresponding to approximately 20 µM. This is only an extremely rough estimate that does not take into account pharmacological parameters. Nevertheless, drug concentrations used in cell culture presumably do not exceed physiological concentrations.

2. There are no experiments to directly prove that the drug impacts VP or VPE biogenesis. To prove this, the authors Will need to label the VP before virus infection and visualise the labelled VP after infection to prove the role of drug in inhibiting VP formation.

Vesicle packets do not represent endogenous organelles, but are induced by dengue virus in a complex interplay of viral and host proteins, as well as DENV RNA. Therefore, VP labeling is just not possible because they only form AFTER infection.

3. If the drug truly inhibits VP biogenesis, wouldn't the cell undergo cell death under drug treatment? Did the investigators test for the health and viability of cells under drug treatment?

It is not clear to us what the reviewer is referring to. Since JNJ-A07 is not cytotoxic at used concentrations (Kaptein et al, Nature, 2021), why should the cell undergo cell death? It's rather to the opposite. Virus infection induces cell death and since JNJ-A07 blocks VPs formation (and thus virus replication), the cell will not die upon drug treatment.

4. Where are the VPs localised? Are they in the ER, or with mitochondria that are known to interact directly with NS4B? The images taken with TEM should be able to shed insights on this.

As reported in the introduction of our manuscript, vesicle packets represent approximately 90 nm invaginations of the rough ER membranes. These structures have already been described in detail in the literature (e.g. Welsch et al, Cell host & microbe, 2009) and our images acquired for this study are consistent with the literature.

5. Did the investigators see dengue virus colocalise with VP in drug and control conditions?

It is not clear to us what the reviewer is referring to. It is not possible to visualize dengue virus particles with VPs because VPs formation is blocked and therefore, all subsequent steps as well, including virus assembly and release. When using the pIRO system, we can induce VPs independent from RNA replication, but in this system virus particles do not form because of the lack of the structural proteins.

6. Using huh-7 and huh-7 subclone does not prove that the experiment is done on 2 cell types. They are essentially same type of cells, unless the investigators can prove that the 2 cells are genetically very different

We agree that these are not different cell types and therefore, had clearly stated Huh7/Lunet is "a subclone of Huh7" and described Huh7 and Huh7/Lunet as "cell lines". We want to point out that in previous work we have tested the antiviral activity of JNJ-A07 in a large panel of cells and different cell types (Kaptein et al, Nature 2021). To avoid confusion, the manuscript has been modified for clarity.

7. If JNJ-A07 inhibits in a similar mechanism as NITD-688, why is it more superior in terms of safety and efficacy profile? Also, are the mechanisms of inhibition similar because JNJ-A07 and NITD-688 share similar structures?

It is likely that the higher potency of JNJ-A07 is due to a higher binding affinity to the target or less side effects. Other pharmacological properties, especially toxicity, are often not related to the primary mechanism-of-action, but to the way in which compounds are metabolized or to off-target effects. In our opinion, this is beyond the scope of this manuscript and very speculative. Therefore, we did not discuss this aspect in the revised manuscript.

8. Note that many other viruses require VP for efficient replication and infection. Does this mean JNJ-A07 would be effective against these viruses as well? The authors should elaborate or at least comment on these in the discussion section

Although VPs formation is a hallmark of many other plus-strand RNA viruses, the viral proteins involved in VPs induction are structurally different and therefore, JNJ-A07 has clear selectivity. In fact, in a previous publication (Kaptein et al, Nature, 2021), we have determined the antiviral activity of JNJ-A07 against a broad panel of RNA and DNA viruses as can be seen in the Rebuttal Table 3 below. Apart from DENV, YFV was most sensitive to JNJ-A07, but antiviral activity was about 1000 times lower

compared to an average value for a DENV isolate. Thus, we classified JNJ-A07 as highly specific for DENV.

[redacted]

Minor comments:

1. Figure 1f. The error bars look overlapping with each other yet the p-values are highly significant. Perhaps the individual points should be plotted to better understand why the comparisons show significance.

We thank the reviewer for this suggestion and have adjusted Figure 1f, as well as Supporting Figure 3a-b accordingly.

Reviewer #3 (Remarks to the Author):

The manuscript from Bartenschlager and colleagues addresses the mechanism by which a known inhibitor, JNJ-A07, affects dengue virus infection. The results complement and expand earlier studies by the group. The data clearly show that the inhibitor interferes with interaction between NS2B/NS3 complex and the intermediate cleavage precursor NS4A-2K-NS4B. The results provide additional insight into understanding processing of the DENV nonstructural polyprotein. Furthermore, the results provide in addition insight into the mechanism of an already identified inhibitor which has the potential to contribute to design of new effective inhibitors.

Comment

1. The data are strong, and experiments were carefully executed but the manuscript overall is very complex with the large amount of supporting data across eight large figures. The need to switch back and forth to the supplemental data is distracting. If only key, necessary selected pieces of data from the supplemental results were incorporated to the extent possible into the primary figures and/or reduction of the large amount of supplemental data would help.

We thank the reviewer for the positive feedback.

As suggested, we have taken the opportunity to move particularly important elements from the supporting figures to the main figures. We also agree that some elements in the supplementary figures were largely redundant. We have reduced these accordingly, without depriving the manuscript of information, as all the data are still available in the source data sheet for a particularly interested readership. Despite the integration of new data, we managed to cut the number of supporting figures from 8 to 5, which greatly improves the reading flow.

REVIEWERS' COMMENTS

Reviewer #1 (Remarks to the Author):

All concerns are addressed.

Reviewer #2 (Remarks to the Author):

The authors have sufficiently addressed my comments. The manuscript is much improved after the reviewers' suggestions.

Reviewer #3 (Remarks to the Author):

The manuscript resubmission from Bartenschlager and colleagues addresses the mechanism by which a known inhibitor, JNJ-A07, affects dengue virus infection. The data remain strong and high quality. The authors have carefully addressed comments from the initially review, added additional information, and made appropriate changes to further strengthen and improve the manuscript. The report provide important insight for an inhibitor and potential for design of new generation inhibitors against a dengue virus, a significantly important pathogen.